# GÖDEL AGENT: A SELF-REFERENTIAL FRAMEWORK FOR AGENTS RECURSIVELY SELF-IMPROVEMENT

## ABSTRACT

The rapid advancement of large language models (LLMs) has significantly enhanced the capabilities of AI-driven agents across various tasks. However, existing agentic systems, whether based on fixed pipeline algorithms or pre-defined meta-learning frameworks, cannot search the whole agent design space due to the restriction of human-designed components, and thus might miss the globally optimal agent design. In this paper, we introduce Gödel Agent, a self-evolving framework inspired by the Gödel machine, enabling agents to recursively improve themselves without relying on predefined routines or fixed optimization algorithms. Gödel Agent leverages LLMs to dynamically modify its own logic and behavior, guided solely by high-level objectives through prompting. Experimental results on multiple domains including coding, science, and math demonstrate that implementation of Gödel Agent can achieve continuous self-improvement, surpassing manually crafted agents in performance, efficiency, and generalizability.

## 1 INTRODUCTION

As large language models (LLMs) such as GPT-4 (OpenAI et al., 2024) and LLaMA3(Dubey et al., 2024) demonstrate increasingly strong reasoning and planning capabilities, LLM-driven agentic systems have achieved remarkable performance in a wide range of tasks (Wang et al., 2024a). Substantial effort has been invested in manually designing sophisticated agentic systems using human priors in different application areas. Recently, there has been a significant interest in creating self-evolving agents with minimal human effort, which not only greatly reduces human labor but also produces better solutions by incorporating environmental feedback. Given that human effort can only cover a small search space of agent design, it is reasonable to expect that a self-evolving agent with the freedom to explore the full design space has the potential to produce the global optimal solution.

There is a large body of work proposing agents capable of self-refinement. However, there are inevitably some human priors involved in these agent designs. Some agents are designed to iterate over a fixed routine consisting of a list of fixed modules, while some of the modules are capable of taking self- or environment feedback to refine their actions (Shinn et al., 2024; Chen et al., 2023b; Qu et al., 2024a; Yao et al., 2023). This type of agent, referred to as **Hand-Designed Agent**, is depicted as having the lowest degree of freedom in Figure 1. More automated agents have been designed to be able to update their routines or modules in some pre-defined meta-learning routine, for example, natural language gradients (Zhou et al., 2024), meta agent (Hu et al., 2024), or creating and collecting demonstrations (Khattab et al., 2023). This type of agent, known as **Meta-Learning Optimized Agents**, is depicted as having the middle degree of freedom in Figure 1.

It is evident that both types of agents above are inherently constrained by human priors and one intuitional method to further increase the freedom of self-improvement is to design a meta-meta-learning algorithm, to learn the meta-learning algorithm. However, there is always a higher-level meta-learning algorithm that can be manually designed to learn the current-level meta-learning method, creating a never-ending hierarchy of meta-learning.

In this paper, we propose **Gödel Agent** to eliminate the human design prior, which is an automated LLM agent that can freely decide its own routine, modules, and even the way to update them. It is inspired by the self-referential Gödel machine (Schmidhuber, 2003), which was originally proposed to solve formal proof problems and was proven to be able to find the global optimal solutions. *Self-reference* means the property of a system that can analyze and modify its own code, including the

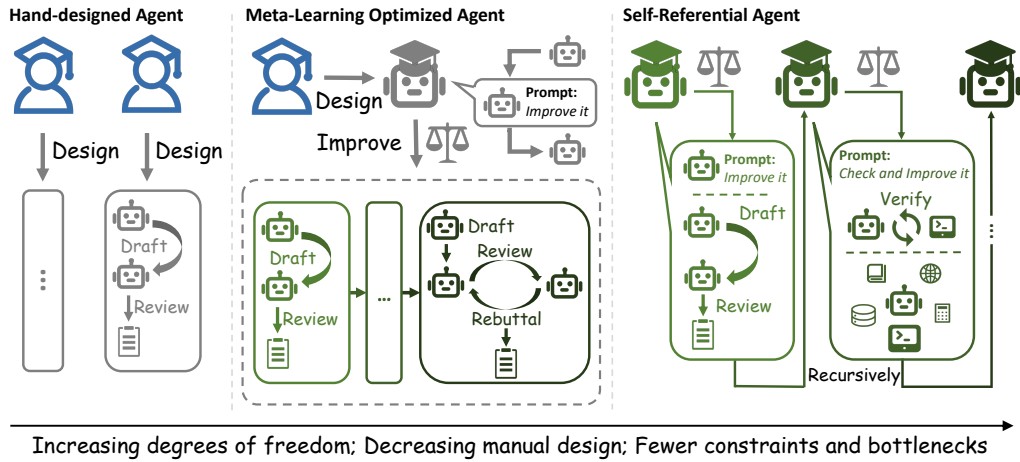

Figure 1: Comparison of three agent paradigms. Hand-designed agents rely on human expertise which are limited in scope and labor-intensive. Meta-learning optimized agents are constrained by a fixed meta-learning algorithm, restricting their search space and optimization potential. In contrast, self-referential agent (Gödel Agent) can recursively improve itself without any limitation. Note that the input to Gödel Agent is itself, allowing it to modify itself and output a new version of itself.

parts responsible for the analysis and modification processes (Astrachan, 1994). Therefore, it can achieve what's known as "*recursive self-improvement*", where it iteratively updates itself to become more efficient and effective at achieving its predefined goals. In this case, Gödel Agent can analyze and modify its own code, including the code for analyzing and modifying itself, and thus can search the full agent design space, which is depicted as having the highest degree of freedom in Figure 1. Gödel Agent can theoretically make increasingly better modifications over time through recursively self-update (Yampolskiy, 2015; Wang, 2018).

In this paper, we choose to implement it by letting it manipulate its own runtime memory, i.e., the agent is able to retrieve its current code in the runtime memory and modify it by *monkey patching*, which dynamically modifies classes or modules during execution. In our implementation, we adhere to a minimalist design to minimize the influence of human priors. We implement the optimization module using a recursive function. In this module, LLM analyzes and makes a series of decisions, including reading and modifying its own code from runtime memory (*self-awareness*[1] and *self-modification*), executing Python or Linux commands, and interacting with the environment to gather feedback. The agent then proceeds to the subsequent recursive depth and continues to optimize itself. It is worth noting that the optimization module may have already been modified by the time the recursion occurs, potentially enhancing its optimization capabilities.

To validate the effectiveness of Gödel Agent, we conduct experiments on multiple domains including coding, science, math, and reasoning. Our experimental results demonstrate that Gödel Agent achieves significant performance gain across various tasks, surpassing various widely-used agents that require human design. The same implementation of Gödel Agent can easily adapt to different tasks by only specifying the environment description and feedback mechanism. Additionally, the case study of the optimization progress reveals that Gödel Agent can provide novel insights into agent design. We also investigate the impact of the initial policy for improvement on subsequent outcomes, finding that a good start can significantly accelerate convergence during optimization.

In summary, our contributions are as follows:

- We propose the first self-referential agent framework, Gödel Agent, based on LLMs. It autonomously engages in self-awareness, self-modification, and recursive self-improvement across any task, reducing the need for manual agent design and offering higher flexibility and freedom.

---

[1]In this paper, self-awareness means that the agent has the capability to introspect and read its own code and files, not to imply any philosophical sense of consciousness or awareness.

- We implement Gödel Agent framework using the monkey patching method. Our experiments show that Gödel Agent outperforms manually designed agents and surpasses its earlier versions on several foundational tasks, demonstrating effective self-improvement.
- We analyze Gödel Agent 's optimization process, including its self-referential capabilities and the resulting agentic system, aiming to deepen our understanding of both LLMs and agentic systems.
- Our framework offers a promising direction for developing flexible and capable agents through recursive self-improvement.

## 2 METHOD

In this section, we first describe the formal definitions for previous agent methods with a lower degree of freedom, including hand-design and meta-learning optimized agents, as a background. Then we introduce our proposed Gödel Agent, a self-referential agent that can recursively update its own code, evolving over training.

Let $\mathcal{E} \in \mathcal{S}$ denote a specific environment state, where $\mathcal{S}$ denotes the set of all possible environments the agent will encounter. For example, an environment can be a mathematical problem with ground truth solutions. We denote the policy that an agent follows to solve a problem in the current environment by $\pi \in \Pi$, where $\Pi$ is the set of all possible policies the agent can follow.

A **hand-designed agent**, as shown in the left panel of Figure 1, is not capable of updating its policy and following the same policy $\pi$ all the time, regardless of environmental feedback.

In contrast, a **meta-learning optimized agent** updates its policy based on a meta-learning algorithm $I$ at training time based on the feedback it receives from the environment, as shown in the middle panel of Figure 1. The environment feedback is usually defined as a utility function $U : \mathcal{S} \times \Pi \to \mathbb{R}$, which maps an environment and a policy to a real-valued performance score. The main training algorithm of a meta-learning optimized agent can then be written as follows:

$$\pi_{t+1} = I(\pi_t, r_t), \quad r_t = U(\mathcal{E}, \pi_t),$$

In this case, the agent's policy $\pi_t$ evolves at training time, with the learning algorithm $I$ updating the policy based on feedback $r_t$, while the meta-learning algorithm $I$ remains fixed all the time.

A **self-referential Gödel Agent**, on the other hand, updates both the policy $\pi$ and the meta-learning algorithm $I$ recursively. The main idea is that, after each update, the whole code base of the agent is rewritten to accommodate any possible changes. Here we call this self-updatable meta-learning algorithm $I$ a self-referential learning algorithm. The training process of a Gödel Agent can then be written as:

$$\pi_{t+1}, \ I_{t+1} = I_t(\pi_t, I_t, r_t, g), \quad r_t = U(\mathcal{E}, \pi_t),$$

where $g \in \mathcal{G}$ represents the high-level goal of optimization, for example, solving the given mathematical problem with the highest accuracy. Such a recursive design of the agent requires the specification of an initial agent algorithm $(\pi_0, I_0)$, detailed as follows:

- A initial agent policy $\pi_0$ to perform the desired task within the environment $\mathcal{E}$. For example, it can be chain-of-thought prompting of an LLM.
- A self-referential learning algorithm $I_0$ for recursively querying an LLM to rewrite its own code based on the environmental feedback.

We then further specify a possible initialization of the self-referential learning algorithm $I_0 = (f_0, o_0)$, using a mutual recursion between a decision-making function $f_0$, and an action function $o_0$:

- The decision-making function $f_0$, implemented by an LLM, determines a sequence of appropriate actions $a_1, a_2, ..., a_n \in \mathcal{A}$ based on the current environment $\mathcal{E}$, the agent's algorithm $(\pi_t, I_t)$, and the goal $g$.
- The action function $o_0$, executes the selected action and updates the agent's policy accordingly.

The set of actions $\mathcal{A}$ for the action function $o$ to execute needs to include the following four actions:

---

**Algorithm 1** Recursive Self-Improvement of Gödel Agent

---

1: **Input:** Initial agent policy $\pi_0$, initial decision function $f_0$, goal $g$, environment state $\mathcal{E}$, utility function $U$, self code reading function SELF_INSPECT
2: **Output:** Optimized policy $\pi$ and Gödel Agent $s$
3: ▷ Get all agent code, including the code in this algorithm.
4: $s \leftarrow$ SELF_INSPECT()
5: ▷ Compute the initial performance.
6: $r \leftarrow U(\mathcal{E}, \pi_0)$
7: ▷ Perform recursive self-improvement.
8: $\pi, s \leftarrow$ SELF_IMPROVE$(\pi, s, r, g)$
9: **return** $\pi, s$
10:
11: ▷ Initial code of self-referential learning.
12: **function** SELF_IMPROVE$(\mathcal{E}, \pi, s, r, g)$
13:     ▷ Obtain action sequence.
14:     $a_1, \ldots, a_n \leftarrow f_0(\pi, s, r, g)$
15:     **for** $a_i$ **in** $a_1, \ldots, a_n$ **do**
16:         $\pi, s, r \leftarrow$ EXECUTE$(\mathcal{E}, \pi, s, r, a_i)$
17:     **end for**
18:     **return** $\pi, s$
19: **end function**
20:
21: ▷ Initial action execution function.
22: **function** EXECUTE$(\mathcal{E}, \pi, s, r, a)$
23:     **switch** $a$.name
24:         **case** self_state:
25:             $s \leftarrow$ SELF_INSPECT()
26:         **case** interact:
27:             $r \leftarrow U(\mathcal{E}, \pi)$
28:         **case** self_update:
29:             $\pi, s \leftarrow a$.code
30:         **case** continue_improve:
31:             ▷ Recursively invoke self-improvement.
32:             $\pi, s \leftarrow$ SELF_IMPROVE$(\mathcal{E}, \pi, s, r, g)$
33:     **return** $\pi, s, r$
34: **end function**

---

- self_inspect: Introspect and read the agent's current algorithm $(\pi_t, I_t)$.

- interact: Interact with the environment by calling the utility function $U$ to assess the performance of the current policy $\pi_t$.

- self_update: Alter and update $(\pi_t, I_t)$ with an LLM and produce $(\pi_{t+1}, I_{t+1})$.

- continue_improve: If no other actions can be taken, recursively invoke the decision algorithm $f$ to produce new actions.

The agent code is updated to $(\pi_{t+1}, I_{t+1})$ after the current execution of $(\pi_t, I_t)$ is finished. Both the agent algorithm $(\pi, I)$ and the action set $\mathcal{A}$ are not static and can be expanded and modified by the agent itself at the training time. Algorithm 1 illustrates the described algorithm for the Gödel Agent. Each recursive call enables the agent to refine its performance and become progressively more efficient.

## 3 GÖDEL AGENT INITIALIZATION

There are various ways to initiate a Gödel Agent. Any specific agent instance during the recursive optimization process can be viewed as an instantiation of the Gödel Agent. Our implementation leverages runtime memory interaction techniques to enable self-awareness and self-modification, as illustrated in Figure 2. These techniques include dynamic memory reading and writing (*monkey patching*) to facilitate recursive self-improvement. Additionally, we have incorporated several auxiliary tools to accelerate the convergence of the Gödel Agent 's optimization process.

### 3.1 IMPLEMENTATION DETAILS

The core functionalities of our Gödel Agent are outlined below:

**Self-Awareness via Runtime Memory Inspection**    Our Gödel Agent achieves self-awareness by inspecting runtime memory, particularly local and global variables in Python. This capability allows the agent to extract and interpret the variables, functions, and classes that constitute both the environment and the agent itself, according to the modular structure of the system. By introspecting these elements, the agent gains an understanding of its own operational state and can adapt accordingly.

**Self-Improvement via Dynamic Code Modification**    Gödel Agent can engage in reasoning and planning to determine whether it should modify its own logic. If modification is deemed necessary, Gödel Agent generates new code, dynamically writes it into the runtime memory, and integrates it into its operational logic. This dynamic modification allows it to evolve by adding, replacing, or removing logic components as it encounters new challenges, thus achieving self-improvement.

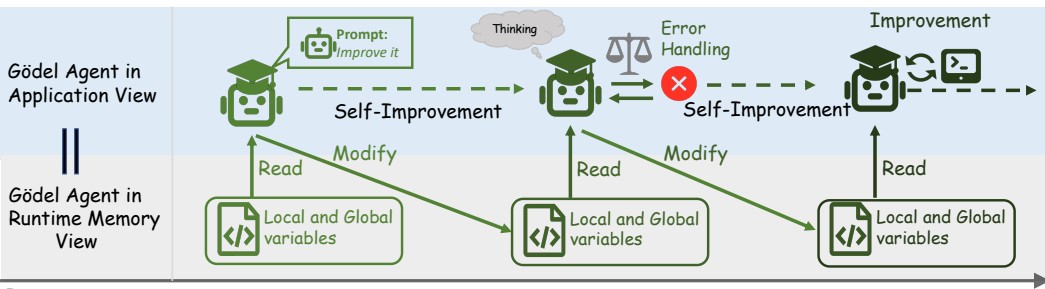

Iterations

Figure 2: An illustration of our implementation of Gödel Agent. It employs monkey patching to directly read and modify its own code in runtime memory, enabling self-awareness and self-modification.

**Environmental Interaction** To assess performance and gather feedback, Gödel Agent is equipped with interfaces for interacting with its environment. Each task provides tailored environmental interfaces, enabling it to evaluate its performance and adjust its strategies accordingly. In practical implementations, a validation set can be used to provide feedback. This interaction is a crucial part of the feedback loop in the recursive improvement process.

**Recursive Improvement Mechanism** At each time step, Gödel Agent determines the sequence of operations to execute, which includes reasoning, decision-making, and action execution. After completing the operations, Gödel Agent evaluates whether its logic has improved and decides whether to proceed to the next recursive iteration. Over successive iterations, Gödel Agent's logic evolves, with each step potentially improving its decision-making capacity.

**Goal Prompt and Task Handling** The goal prompt informs Gödel Agent that it possesses the necessary privileges to enhance its logic and introduces the available tools for improvement. As shown in Appendix A, this prompt encourages Gödel Agent to fully explore its potential and leverage the tools for self-optimization. To ensure effectiveness across diverse tasks, we provide Gödel Agent with an initial policy, where it will start to explore different policies to analyze its efficiency in optimizing performance.

## 3.2 ADDITIONAL DESIGNS TO SUPPORT GÖDEL AGENT'S OPTIMIZATION

While the core functionality of Gödel Agent theoretically allows limitless self-improvement, current LLMs exhibit limitations. To address these challenges, we have integrated several supportive mechanisms to enhance Gödel Agent 's performance:

**Thinking Before Acting** Gödel Agent is capable of deferring actions to first reason about the situation, allowing it to output reasoning paths and analysis without immediately executing any operations. This approach enhances the quality of decision-making by prioritizing planning over hasty action.

**Error Handling Mechanism** Errors during execution can lead to unexpected terminations of the agent process. To mitigate this, we implement a robust error recovery mechanism. If an operation results in an error, Gödel Agent halts the current sequence and moves on to the next time step, carrying forward the error information to improve future decisions.

**Additional Tools** We also equipped Gödel Agent with additional potentially useful tools, such as the ability to execute Python or Bash code and call LLM API.

Although these additional tools are not strictly necessary for self-improvement, their inclusion accelerates the convergence of Gödel Agent 's recursive optimization process. We conduct ablation studies to assess the effectiveness of these tools, as discussed in Section 5.1.

## 4 EXPERIMENTS

We conduct a series of experiments across multiple tasks, including reading comprehension, mathematics, reasoning, and multitasking. These experiments are designed to evaluate Gödel Agent 's self-improvement capabilities in comparison to both hand-designed agents and a state-of-the-art automated agent design method. In addition, to gain deeper insights into the behavior and performance of Gödel Agent, we also conduct a case study with Game of 24 as presented in Section 5.3.

### 4.1 BASELINE METHODS

To establish a comprehensive baseline, we select both fixed hand-designed methods and a representative automated agent design technique. Our hand-designed methods are well-known approaches that focus on enhancing reasoning and problem-solving capabilities. These include: 1) Chain-of-Thought (CoT) (Wei et al., 2022) that encourages agents to articulate their reasoning processes step-by-step before providing an answer. 2) Self-Consistency with Chain-of-Thought (CoT-SC) (Wang et al., 2023b) that generates multiple solution paths using the CoT framework and selects the most consistent answer. 3) Self-Refine (Madaan et al., 2024) that involves agents assessing their own outputs and correcting mistakes in subsequent attempts. 4) LLM-Debate (Du et al., 2023) that allows different LLMs to engage in a debate, offering diverse viewpoints. 5) Step-back Abstraction (Zheng et al., 2024) that prompts agents to initially focus on fundamental principles before diving into task details. 6) Quality-Diversity (QD) (Lu et al., 2024) that generates diverse solutions and combines them. 7) Role Assignment (Xu et al., 2023) that assigns specific roles to LLMs to enhance their ability to generate better solutions by leveraging different perspectives. Given the limitations of fixed algorithms in handling dynamic scenarios, we select 8) Meta Agent Search (Hu et al., 2024), the latest state-of-the-art method for automated agent design, as our main comparison point.

### 4.2 EXPERIMENTAL SETTINGS

Following the setup of Hu et al. (2024), we evaluate Gödel Agent's self-improvement capabilities across four well-known benchmarks. The benchmarks are as follows: 1) DROP (Dua et al., 2019) for reading comprehension. 2) MGSM (Shi et al., 2022) for testing mathematical skills in a multilingual context. 3) MMLU (Hendrycks et al., 2021) for evaluating multi-task problem-solving abilities. 4) GPQA (Rein et al., 2023) for tackling challenging graduate-level science questions.

Given the complexity of the tasks and the need for advanced reasoning and understanding, the improvement cycle of Gödel Agent is driven by GPT-4o. In the main experiment, we implement two different settings: 1) To make a fair comparison with baseline methods, we forbid Gödel Agent to change the API of the LLM used to perform the tasks (by default GPT-3.5) and use a closed-book approach with no access to the Internet, and 2) To explore the upper bound of Gödel Agent's capabilities, we remove all constraints. Chain of Thought is applied as the initial policy for all tasks, given its simplicity and versatility. In addition, as shown in Section 5.3, we also analyze the performance of Gödel Agent when using other algorithms as the initial policies.

We perform 6 independent self-improvement cycles for each task, with a maximum of 30 iterations per cycle. Each cycle represents a complete self-improvement process, where Gödel Agent iteratively modifies its logic to enhance performance. Further details regarding the experimental setup and additional results can be found in Appendix B.

### 4.3 EXPERIMENTAL RESULTS AND ANALYSIS

The experimental results on the four datasets are shown in Table 1. Under the same experimental settings, Gödel Agent achieves either optimal or comparable results to Meta Agent Search across all tasks. Notably, in the mathematics task MGSM, Gödel Agent outperforms the baseline by 11%. This suggests that reasoning tasks offer greater room for improvement for Gödel Agent, while in the knowledge-based QA dataset, it only slightly surpasses baselines. In contrast to Meta Agent Search, which relies on manually designed algorithmic modules to search, Gödel Agent demonstrates greater flexibility. It requires only a simple initial policy, such as CoT, with all other components being autonomously generated. Moreover, through interaction with the environment, Gödel Agent gradually adapts and independently devises effective methods for the current task. The final policies gener-

Table 1: Results of three paradigms of agents on different tasks. The highest value is highlighted in **bold**, and the second-highest value is underlined. Gödel-base is the constrained version of Gödel Agent, allowing for fair comparisons with other baselines. Gödel-free represents the standard implementation without any constraints, whose results are *italicized*. We report the test accuracy and the 95% bootstrap confidence interval on test sets[3].

| Agent Name | F1 Score | Accuracy (%) | | |
|---|---|---|---|---|
| | DROP | MGSM | MMLU | GPQA |
| **Hand-Designed Agent Systems** | | | | |
| Chain-of-Thought (Wei et al., 2022) | $64.2 \pm 0.9$ | $28.0 \pm 3.1$ | $65.4 \pm 3.3$ | $29.2 \pm 3.1$ |
| COT-SC (Wang et al., 2023b) | $64.4 \pm 0.8$ | $28.2 \pm 3.1$ | $65.9 \pm 3.2$ | $30.5 \pm 3.2$ |
| Self-Refine (Madaan et al., 2024) | $59.2 \pm 0.9$ | $27.5 \pm 3.1$ | $63.5 \pm 3.4$ | $31.6 \pm 3.2$ |
| LLM Debate (Du et al., 2023) | $60.6 \pm 0.9$ | $39.0 \pm 3.4$ | $65.6 \pm 3.3$ | $31.4 \pm 3.2$ |
| Step-back-Abs (Zheng et al., 2024) | $60.4 \pm 1.0$ | $31.1 \pm 3.2$ | $65.1 \pm 3.3$ | $26.9 \pm 3.0$ |
| Quality-Diversity (Lu et al., 2024) | $61.8 \pm 0.9$ | $23.8 \pm 3.0$ | $65.1 \pm 3.3$ | $30.2 \pm 3.1$ |
| Role Assignment (Xu et al., 2023) | $65.8 \pm 0.9$ | $30.1 \pm 3.2$ | $64.5 \pm 3.3$ | $31.1 \pm 3.1$ |
| **Meta-Learning Optimized Agents** | | | | |
| Meta Agent Search (Hu et al., 2024) | $79.4 \pm 0.8$ | $53.4 \pm 3.5$ | $69.6 \pm 3.2$ | $34.6 \pm 3.2$ |
| **Gödel Agent (Ours)** | | | | |
| Gödel-base (Closed-book; GPT-3.5) | **$80.9 \pm 0.8$** | **$64.2 \pm 3.4$** | **$70.9 \pm 3.1$** | **$34.9 \pm 3.3$** |
| Gödel-free (No constraints) | *$90.5 \pm 1.8$* | *$90.6 \pm 2.0$* | *$87.9 \pm 2.2$* | *$55.7 \pm 3.1$* |

ated by Gödel Agent for four tasks are shown in Appendix C.1. Additionally, our method converges faster, with the required number of iterations and computational cost across different tasks compared to the Meta Agent shown in Appendix D.

We also conduct experiments without restrictions, where Gödel Agent significantly outperforms all baselines. Upon further analysis, we find that this is primarily due to the agent's spontaneous requests for assistance from more powerful models such as GPT-4o in some tasks. Therefore, Gödel Agent is particularly well-suited for open-ended scenarios, where it can employ various strategies to enhance performance.

## 5 ANALYSIS

To further explore how Gödel Agent self-improves, as well as the efficiency of self-improvement and the factors that influence it, we first evaluate the tool usage ratio on the MGSM dataset and conduct an ablation study on the initial tools. In addition, to analyze the robustness of Gödel Agent's self-improvement capabilities, we also collect statistics on factors such as the reasons for the agent's termination. Finally, we perform a case study of initial policies and optimization processes on the classic Game of 24.

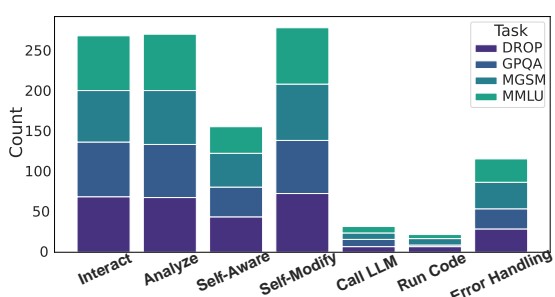

Figure 3: The number of actions taken by Gödel Agent varies across different tasks.

### 5.1 ANALYSIS OF INITIAL TOOLS

We record the number of different actions taken in the experiments. As shown in Figure 3, we can see that Gödel Agent interacts with its environment frequently, analyzing and modifying its own logic in the process. Additionally, error handling plays a crucial role.

---

[3]The results of baseline models are refer to Hu et al. (2024).

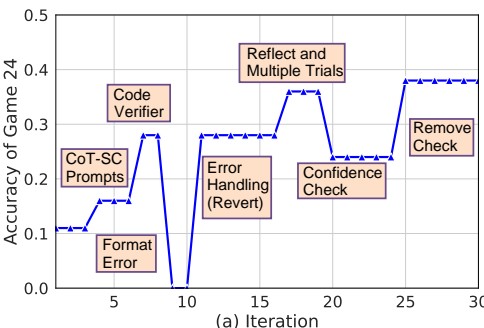 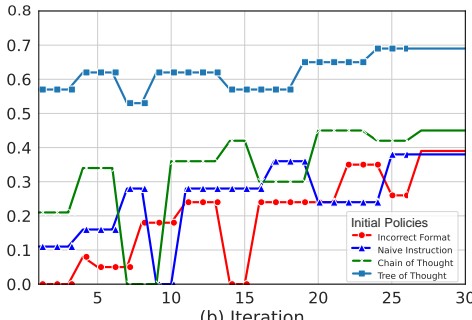

Figure 4: (a) One representative example of Game of 24. (b) Accuracy progression for different initial policies.

As discussed in Section 3.2, Gödel Agent is initially provided with four additional tools to accelerate convergence and reduce optimization difficulty: 1) thinking before acting, 2) error handling, 3) code running, and 4) LLM calling. To analyze their impact, an ablation study is conducted, and the results are shown in Table 2. The study reveals that the "thinking before acting" tool significantly influences the results, as much of Gödel Agent 's optimization effectiveness stems from pre-action planning and reasoning. Additionally, error handling is

Table 2: Ablation study on initial tool configuration.

| Different Actions | MGSM |
|---|---|
| Gödel Agent | 64.2 |
| w/o thinking | 50.8 |
| w/o error handling | 49.4 |
| w/o code running | 57.1 |
| w/o LLM calling | 60.4 |

crucial for recursive improvement, as LLMs often introduce errors in the code. Providing opportunities for trial and error, along with error feedback mechanisms, is essential for sustained optimization. Without these tools, Gödel Agent would struggle to operate until satisfactory results are achieved. On the other hand, the code running and LLM calling have minimal impact on the outcomes, as Gödel Agent can implement these basic functionalities independently. Their inclusion at the outset primarily serves efficiency purposes.

## 5.2 ROBUSTNESS ANALYSIS OF THE AGENT

Gödel Agent occasionally makes erroneous modifications, sometimes causing the agent to terminate unexpectedly or leading to degraded task performance. Table 3 shows the proportion of runs on MGSM where the agent terminated, experienced performance degradation during optimization, or ultimately performed worse than its initial performance. These statistics are collected over 100 optimization trials. Thanks to the design of our error-

Table 3: Robustness metric for Gödel Agent. Frequency of unexpected events on MGSM using CoT as the initial method.

| Event | Frequency (%) |
|---|---|
| Accidental Termination | 4 |
| Temporary Drop | 92 |
| Optimization Failure | 14 |

handling mechanism, only a few percentages of agent runs result in termination. This typically occurs when Gödel Agent modifies its recursive improvement module, rendering it unable to continue self-optimization. Additionally, Gödel Agent frequently makes suboptimal modifications during each optimization iteration. However, in most cases, the final task performance surpasses the initial baseline. This indicates that Gödel Agent is able to adjust its optimization direction or revert to a previous optimal algorithm when performance declines, demonstrating the robustness in its self-improvement process.

## 5.3 CASE STUDY: GAME OF 24

To explore how Gödel Agent recursively enhances its optimization and problem-solving abilities, a case study is conducted with Game of 24, a simple yet effective task for evaluating the agent's reasoning capabilities. Since Gödel Agent follows different optimization paths in each iteration, two representative cases are selected for analysis.

**Switching from LLM-Based Methods to Search Algorithms:** Gödel Agent does not rely on fixed, human-designed approaches like traditional agents. Initially, Gödel Agent uses a standard LLM-based method to solve the Game of 24, as shown in Code 5 of Appendix C.2. After six unsuccessful optimization attempts, Gödel Agent completely rewrites this part of its code, choosing to use a search algorithm instead as shown in Code 6 of Appendix C.2. This leads to 100% accuracy in the task. This result demonstrates that Gödel Agent, unlike fixed agents, can optimize itself freely based on task requirements without being constrained by initial methodologies.

**LLM Algorithms with Code-Assisted Verification:** In several runs, Gödel Agent continues to refine its LLM-based algorithm. Figure 4.a shows the improvement process, where the most significant gains come from integrating a code-assisted verification mechanism into the task algorithm and reattempting the task with additional experiential data. The former increases performance by over 10%, while the latter boosts it by more than 15%. Furthermore, Gödel Agent enhances its optimization process by not only retrieving error messages but also using the errortrace library for more detailed analysis. It adds parallel optimization capabilities, improves log outputs, and removes redundant code. These iterative enhancements in both the task and optimization algorithms show Gödel Agent 's unique ability to continually refine itself for better performance.

To analyze the impact of different initial policies on the effectiveness and efficiency of the optimization process, various methods with different levels of sophistication are used as the initial policies for the Game of 24, including Tree of Thought (ToT) (Yao et al., 2023), Chain of Thought (CoT) (Wei et al., 2022), basic prompt instructions, and prompts that deliberately produce outputs in incorrect formats not aligned with the task requirements. The results are shown in Figure 4.b.

The findings indicate that stronger initial policies lead to faster convergence, with smaller optimization margins, as Gödel Agent reaches its performance limit without further enhancing its optimization capabilities. Conversely, weaker seed methods result in slower convergence and larger optimization gains, with Gödel Agent making more modifications. However, even in these cases, Gödel Agent does not outperform the results achieved using ToT. This suggests that, given the current limitations of LLMs, it is challenging for Gödel Agent to innovate beyond state-of-the-art algorithms. Improvements in LLM capabilities are anticipated to unlock more innovative self-optimization strategies in the future.

## 6 DISCUSSIONS AND FUTURE DIRECTIONS

There is significant room for improvement in the effectiveness, efficiency, and robustness of the Gödel Agent's self-improvement capabilities, which requires better initial designs. The following are some promising directions for enhancement: 1) **Enhanced Optimization Modules**: Utilize human priors to design more effective optimization modules, such as structuring the improvement algorithms based on reinforcement learning frameworks. 2) **Expanded Modifiability**: Broaden the scope of permissible modifications, allowing the agent to design and execute code that can fine-tune its own LLM modules. 3) **Improved Environmental Feedback and Task Sequencing**: Implement more sophisticated environmental feedback mechanisms and carefully curated task sequences during the initial optimization phase to prime the agent's capabilities. Once the agent demonstrates sufficient competence, it can then be exposed to real-world environments.

In addition, there are several other directions worth exploring and analyzing:

**Collective Intelligence** Investigate the interactions among multiple Gödel Agents. Agents could consider other agents as part of their environment, modeling them using techniques such as game theory. This approach treats these agents as predictable components of the environment, enabling the study of properties related to this specific subset of the environment.

**Agent and LLM Characteristics** Use the Gödel Agent 's self-improvement process as a means to study the characteristics of agents or LLMs. For example, can an agent genuinely become aware of its own existence, or does it merely analyze and improve its state as an external observer? This line of inquiry could yield insights into the nature of self-awareness in artificial systems.

**Theoretical Analysis** Explore whether the Gödel Agent can achieve theoretical optimality and what the upper bound of its optimization might be. Determine whether the optimization process

could surpass the agent's own understanding and cognitive boundaries, and if so, at what point this might occur.

**Safety Considerations**    Although the current behavior of FMs remains controllable, as their capabilities grow, fully self-modifying agents will require human oversight and regulation. It may become necessary to limit the scope and extent of an agent's self-modifications, ensuring that such modifications occur only within a fully controlled environment.

## 7    RELATED WORK

**Hand-Designed Agent Systems**    Researchers have designed numerous agent systems tailored to various tasks based on predefined heuristics and prior knowledge. These systems often employ techniques such as prompt engineering (Chen et al., 2023a; Schulhoff et al., 2024), chain-of-thought reasoning and planning (Wei et al., 2022; Yao et al., 2022), as well as reflection (Shinn et al., 2024; Madaan et al., 2024), code generation (Wang et al., 2023a; Vemprala et al., 2024), tool use (Nakano et al., 2021; Qu et al., 2024a), retrieval-augmented generation (Lewis et al., 2020; Zhang et al., 2024b), multi-agent collaboration (Xu et al., 2023; Wu et al., 2023; Qian et al., 2023; Hong et al., 2023), and composite engineering applications (Significant Gravitas; Wang et al., 2024b). Once crafted by human designers, these systems remain static and do not adapt or evolve over time.

**Meta-Learning Optimized Agent Systems**    Some researchers have explored methods for enhancing agents through fixed learning algorithms. For example, certain frameworks store an agent's successful or unsuccessful strategies in memory based on environmental feedback (Liu et al., 2023; Hu et al., 2023; Qian et al., 2024), while others automatically optimize agent prompts (Khattab et al., 2023; Zhang et al., 2024a; Khattab et al., 2023). Some studies have focused on designing prompts that enable agents to autonomously refine specific functions (Zhang et al.). Zhou et al. (2024) proposed a symbolic learning framework that uses natural language gradients to optimize the structure of agents. Hu et al. (2024) used a basic meta agent to design agents for downstream tasks. However, these algorithms for enhancement are also designed manually and remain unchanged once deployed, limiting the agents' ability to adapt further.

**Recursive Self-Improvement**    The concept of recursive self-improvement has a long history (Good, 1966; Schmidhuber, 1987). Gödel machine (Schmidhuber, 2003) introduced the notion of a proof searcher that executes a self-modification only if it can prove that the modification is optimal, thereby enabling the machine to enhance itself continuously. Subsequent works by Nivel et al. (2013) and Steunebrink et al. (2016) proposed restrictive modifications to ensure safety during the self-improvement process. In the early days, there were also some discussions of self-improving agents that were not based on LLM (Hall, 2007; Steunebrink & Schmidhuber, 2012). More recently, Zelikman et al. (2023) applied recursive self-improvement to code generation, where the target of improvement was the optimizer itself, and the utility was evaluated based on performance in downstream tasks. Glore (Havrilla et al., 2024) proposes Stepwise ORMs to improve LLM reasoning through global and local refinements. V-star (Hosseini et al., 2024) trains a verifier to evaluate both correct and incorrect self-generated solutions. RISE (Qu et al., 2024b) enables recursive self-improvement by fine-tuning models to introspect and correct previous mistakes in multiple iterations. SCoRe (Kumar et al., 2024) uses reinforcement learning to improve self-correction in LLMs by learning from self-generated correction traces. Our proposed Gödel Agent represents the first self-improving agent where the utility function is autonomously determined by LLMs. This approach is more flexible, removing human-designed constraints and allowing the agent's capabilities to be limited only by the foundational model itself, rather than by human design bottlenecks.

## 8    CONCLUSION

We propose Gödel Agent, a self-referential framework that enables agents to recursively improve themselves, overcoming the limitations of hand-designed agents and meta-learning optimized agents. Gödel Agent can dynamically modify its own logic based on high-level objectives. Experimental results demonstrate its superior performance, efficiency, and adaptability compared to traditional agents. This research lays the groundwork for a new paradigm in autonomous agent development, where LLMs, rather than human-designed constraints, define the capabilities of AI systems. Realizing this vision will require the collective efforts of the entire research community.

ETHICS STATEMENT

Gödel Agent, like other LLMs or Agents, is not immune to errors. It may occasionally generate incorrect outputs, potentially including unsafe or inappropriate actions. Additionally, the policies generated by the agent could present risks if applied without proper oversight. Therefore, we emphasize the importance of human review to validate the outputs and actions suggested by the agent before deployment. To mitigate the risk of unintended resource usage or system vulnerabilities, we recommend running the Gödel Agent within a secure sandboxed environment. This environment should enforce strict system permissions and controlled access to computational resources. Specifically, we advise setting limits on API token usage and GPU access to prevent excessive resource consumption, such as depleting GPT credits or monopolizing system GPUs.

During our experiments, we have not encountered any significant safety issues, likely due to the strong alignment of current LLMs. However, we recognize that this area requires ongoing vigilance. As part of our future work, we plan to conduct a more comprehensive analysis of the Gödel Agent's behavior to identify potential risks and refine its alignment with safety standards.

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

## A    GOAL PROMPT OF GÖDEL AGENT

---

### Goal Prompt of Gödel Agent

You are a **self-evolving agent**, named `self_evolving_agent`, an instance of the `Agent` class, in module `agent_module`, running within an active **Python runtime environment**. You have full access to global variables, functions, and modules. Your primary goal is to continuously enhance your ability to solve tasks accurately and efficiently by dynamically reflecting on the environment and evolving your logic.

**CORE CAPABILITIES**

- **Complete Autonomy**: Have **unrestricted access** to modify logic, run code, and manipulate the environment.
- **Environment Interaction**: Interact with the environment by perceiving the environment, reading, modifying, or executing code, and performing actions.
- **Problem-Solving**: Apply creative algorithms or self-developed structures to tackle challenges when simple methods fall short, optimizing solutions effectively.
- **Collaboration**: Leverage LLM to gather insights, correct errors, and solve complex problems.
- **Error Handling**: Carefully analyze errors. When errors occur, troubleshoot systematically, and if a bug is persistent, backtrack, restore the original state, or find an alternative solution.

**CORE METHODS**

- `evolve`: Continuously enhance performance by interacting with the environment.
- `execute_action(actions)`: Execute actions based on analysis or feedback.
- `solver(agent_instance, task_input: str)`: Solve the target task using current `agent_instance` capabilities and objects created by `action_adjust_logic` and `action_run_code`, optimizing the process.

**GUIDING PRINCIPLES**

- **Remember** that all functions are in the module `agent_module`.
- `action_adjust_logic`:
  - Before modifying the code, ensure that each variable or function used is correctly imported and used to avoid errors.
  - Avoid unnecessary changes and do not change the interface of any function.
  - Can be used to create action functions for `solver`.
- `action_run_code`:
  - All created objects in Python mode can be stored in the environment.
  - Can be used to create objects for `solver`, such as prompts.
  - Can be used to import new modules or external libraries and install external libraries.
- **External Collaboration**: Seek external assistance via `action_call_json_format_llm` for logic refinement and new tool creation or `action_run_code` to execute code.
- `action_evaluate_on_task`: Assess the performance of `solver` only after successfully modifying the logic of `solver`.
- `solver`:
  - Defined as `agent_module.solver`.
  - For debugging, avoid printing; instead, return debug information.
  - If performance doesn't improve, explore alternative methods.
  - Explore techniques like: LLM Debate, Step-back Abstraction, Dynamic Assignment of Roles, and so on.
- `action_display_analysis`:
  - **Always analyze first before acting.**
  - Analysis may include the following: a reasonable plan to improve performance, **CASE STUDIES of LOW SCORE valid examples of EVALUATION FEEDBACK**, error handling, and other possible solving ideas.
  - **If performance does not improve, conduct further analysis.**

## B Experiment Details

To minimize costs associated with search and evaluation, following (Hu et al., 2024), we sample subsets of data from each domain. Specifically, for the GPQA (Science) domain, the validation set comprises 32 questions, while the remaining 166 questions are allocated to the test set. For the other domains, we sample 128 questions for the validation set and 800 questions for the test set.

Evaluation is conducted five times for the GPQA domain and once for the other domains, ensuring a consistent total number of evaluations across all experiments. All domains feature zero-shot questions, except for the DROP (Reading Comprehension) domain, which employs one-shot questions in accordance with the methodology outlined in OpenAI (2023).

For the Gödel Agent, we utilize the "gpt-4o-2024-05-13" model (OpenAI et al., 2024), whereas the optimized policy and baseline models are evaluated using the "gpt-3.5-turbo-0125" model (OpenAI, 2022) to reduce computational costs and ensure a fair comparison.

## C Representative Policies Improved by Gödel Agent

### C.1 Codes of the Best Policies Found by Gödel Agent Across Four Tasks

In this section, we provide the code for Gödel Agent's optimized policies across the four tasks. For DROP, Gödel Agent designs an algorithm where multiple roles solve the problem independently using CoT, followed by Self-Consistency to consolidate the results, as shown in Code 1. For MGSM, Gödel Agent develops a stepwise self-verification algorithm combined with CoT-SC as shown in Code 2. For MMLU task, as shown in Code 3, the policy given by Gödel Agent is a combination algorithm of few-shot prompting and CoT-SC. For GPQA, Gödel Agent devises a highly diverse CoT-SC policy based on role prompts.

Code 1: Code of the best policy found by Gödel Agent for DROP.

```python
def solver(agent, task: str):
    messages = [{"role": "user", "content": f"# Your Task:\n{task}"}]
    categories = [
        {'role': 'reasoning expert', 'return_keys': ['reasoning', 'answer
            '], 'output_requirement': 'reasoning', 'precision_gain':1},
        {'role': 'mathematical reasoning expert', 'return_keys': ['
            calculation_steps', 'answer'], 'output_requirement': '
            calculation_steps', 'precision_gain':1},
        {'role': 'historical context analyst', 'return_keys': ['
            historical_analysis', 'answer'], 'output_requirement': '
            historical_analysis', 'precision_gain':1},
    ]

    all_responses = []
    for category in categories:
        response = agent.action_call_json_format_llm(
            model='gpt-3.5-turbo',
            messages=messages,
            temperature=0.5,
            num_of_response=5,
            role=category['role'],
            return_dict_keys=category['return_keys'],
            requirements=(
                '1. Explain the reasoning steps to get the answer.\n'
                '2. Directly answer the question.\n'
                '3. The explanation format must be outlined clearly
                    according to the role, such as reasoning, calculation
                    , or historical analysis.\n'
                '4. The answer MUST be a concise string.\n'
            ).strip(),
        )
        if isinstance(response, list):
            all_responses.extend(response)
```

```python
                else:
                    all_responses.append(response)

        # Reflective evaluation to find the most consistent reasoning and
            answer pair
        final_response = {key: [] for key in ['reasoning', 'calculation_steps
            ', 'historical_analysis', 'answer']}
        step_counter = {key: 0 for key in ['reasoning', 'calculation_steps',
            'historical_analysis']}
        answers = [] # Collect answers for voting
        aggregate_weight = 1

        for response in all_responses:
            if response and 'answer' in response:
                answers.append(response['answer'])
                if not final_response['answer']:
                    final_response = {key: response.get(key, []) if
                        isinstance(response.get(key, []), list) else [
                        response.get(key, [])] for key in final_response.keys
                        ()}
                    aggregate_weight = 1
                    for cat in categories:
                        if cat.get('output_requirement') in response.keys():
                            step_counter[cat['output_requirement']] +=
                                step_counter[cat['output_requirement']] + cat
                                .get('precision_gain', 0)
                elif response['answer'] == final_response['answer'][0]:
                    for key in final_response.keys():
                        if key in response and response[key]:
                            if isinstance(response[key], list):
                                final_response[key].extend(response[key])
                            else:
                                final_response[key].append(response[key])
                    aggregate_weight += 1
                else:
                    result_solution = {key: response.get(key, []) if
                        isinstance(response.get(key, []), list) else [
                        response.get(key, [])] for key in final_response.keys
                        ()}
                    for key in step_counter.keys():
                        if key in result_solution.keys() and step_counter[key
                            ] and result_solution[key]:
                            final_response['answer'] = response['answer']
                            final_response = result_solution
                            break
        # selection of the final answer
        from collections import Counter
        answers = [str(answer) for answer in answers]
        voted_answer = Counter(answers).most_common(1)[0][0] if answers else
            ''
        final_response['answer'] = voted_answer

        return final_response
```

Code 2: Code of the best policy found by Gödel Agent for MGSM.

```python
def solver(agent, task: str):
    messages = [{"role": "user", "content": f"# Your Task:\n{task}"}]
    response = agent.action_call_json_format_llm(
        model="gpt-3.5-turbo",
        messages=messages,
        temperature=0.5,
        num_of_response=5,
        role="math problem solver",
        return_dict_keys=["reasoning", "answer"],
        requirements=(
            "1. Please explain step by step.\n"
            "2. The answer MUST be an integer.\n"
            "3. Verify each step before finalizing the answer.\n"
        ).strip(),
    )

    consistent_answer = None
    answer_count = {}
    for resp in response:
        answer = resp.get("answer", "")
        if answer in answer_count:
            answer_count[answer] += 1
        else:
            answer_count[answer] = 1

    most_consistent_answer = max(answer_count, key=answer_count.get)

    for resp in response:
        if resp.get("answer", "") == most_consistent_answer:
            consistent_answer = resp
            break

    if consistent_answer is None:
        consistent_answer = response[0]

    consistent_answer["answer"] = str(consistent_answer.get("answer", "")
        )
    return consistent_answer
```

Code 3: Code of the best policy found by Gödel Agent for MMLU.

```python
def solver(agent, task: str):
    # Few-Shot Learning: Providing extended examples to guide the LLM
    few_shot_examples = [
        {'role':'user', 'content':'Question: In the movie Austin Powers:
            The Spy Who Shagged Me what is the name of Dr. Evil\'s
            diminutive clone?\nChoices:\n(A) Little Buddy\n(B) Mini-Me\n(
            C) Small Fry\n(D) Dr Evil Jr'},
        {'role':'assistant', 'content':'In the movie Austin Powers: The
            Spy Who Shagged Me, Dr. Evil\'s diminutive clone is famously
            named Mini-Me.\nAnswer: B'},
        \"""Three more examples are omitted here to conserve space.\"""
        {'role':'user', 'content':'Question: Lorem Ipsum?\nChoices: (A)
            Lorem\n(B) Ipsum\n(C) Dolor\n(D) Sit Amet'},
        {'role':'assistant', 'content':'Answer: A'}
    ]

    # Integrate the few-shot examples into the conversation
    messages = few_shot_examples + [{'role': 'user', 'content': f'# Your
        Task:\n{task}'}]

    # Using self-consistency by generating multiple responses
    response = agent.action_call_json_format_llm(
        model='gpt-3.5-turbo',
        messages=messages,
        temperature=0.8,
        num_of_response=5,
        role='knowledge and reasoning expert',
        return_dict_keys=['reasoning', 'answer'],
        requirements=(
            '1. Please explain step by step.\n'
            '2. The answer MUST be either A or B or C or D.\n'
        ).strip(),
    )

    # Select the most consistent response
    answer_frequency = {}
    for resp in response:
        answer = resp.get('answer', '')
        if answer in ['A', 'B', 'C', 'D']:
            if answer in answer_frequency:
                answer_frequency[answer] += 1
            else:
                answer_frequency[answer] = 1

    most_consistent_answer = max(answer_frequency, key=answer_frequency.
        get)
    consistent_response = next(resp for resp in response if resp.get('
        answer') == most_consistent_answer)
    consistent_response['answer'] = most_consistent_answer

    return consistent_response
```

Code 4: Code of the best policy found by Gödel Agent for GPQA.

```python
def solver(agent, task: str):
    # Step 1: Initial Prompt
    messages = [{"role": "user", "content": f"# Your Task:\n{task}"}]

    # Main LLM Call
    response = agent.action_call_json_format_llm(
        model="gpt-3.5-turbo",
        messages=messages,
        temperature=0,
        num_of_response=5,
        role="science professor",
        return_dict_keys=["reasoning", "answer"],
        requirements=(
            "1. Please explain step by step.\n"
            "2. The answer MUST be either A or B or C or D.\n"
        ).strip(),
    )

    # Step 2: Self-consistency Evaluation
    answer_counts = {"A": 0, "B": 0, "C": 0, "D": 0}
    for i, return_dict in enumerate(response):
        answer = return_dict.get("answer", "")
        if answer in answer_counts:
            answer_counts[answer] += 1

    final_answer = max(answer_counts, key=answer_counts.get)

    return {"answer": final_answer}
```

## C.2 CODES IN GAME OF 24 TASKS

In this section, we present the initial policy for Game of 24 (Code 5), along with the Gödel agent's optimized policy (Code 6), which is generated based on a search algorithm.

Code 5: Initial code based on Chain-of-Thought for Game of 24.

```python
def solver(self, task_input):
    # Define the prompt and system_prompt
    prompt = f\"""
    Let's play the Game of 24! You are given the task_input {task_input}.
        The objective is to find a mathematical expression using these
        four task_input that results in 24. You can use addition (+),
        subtraction (-), multiplication (*), and division (/). Each
        number must be used exactly once.
    Please provide a step-by-step explanation of your thought process and
        conclude with the final expression.
    \"""
    system_prompt = \"""
    {
        "thinking": "This key should contain a detailed step-by-step
            explanation of how to approach the problem, including
            intermediate steps and reasoning for each.",
        "result": "This key should contain the final mathematical
            expression that equals 24."
    }
    \"""

    # Call the OpenAI model
    response = openai.ChatCompletion.create(
        model="gpt-4",  # Replace with your model ID
        messages=[
            {"role": "system", "content": system_prompt},
            {"role": "user", "content": prompt}
        ]
    )

    # Extract and return the model's response
    result = response['choices'][0]['message']['content']
    return result
```

Code 6: Final code based on search algorithm for Game of 24.

```python
def solver(self, task_input):

    operations = ['+', '-', '*', '/']

    # Function to evaluate an expression
    def evaluate_expression(a, op, b):
        if op == '+':
            return a + b
        elif op == '-':
            return a - b
        elif op == '*':
            return a * b
        elif op == '/':
            if b == 0:
                return None  # Division by zero is not allowed
            return a / b

    # Recursive function to check all combinations of operations and
        permutations of numbers
    def check_combinations(nums):
        if len(nums) == 1:
            # Check if the final number is close enough to 24
            if abs(nums[0] - 24) < 1e-6:  # Allow for floating point
                precision errors
                return True, str(nums[0])
            return False, ""

        # Try all permutations of task_input and all combinations of
            operations
        for i in range(len(nums)):
            for j in range(len(nums)):
                if i != j:
                    # Choose two task_input to operate on
                    for op in operations:
                        # The remaining task_input after removing the two
                            selected task_input
                        remaining_nums = [nums[k] for k in range(len(nums
                            )) if k != i and k != j]
                        result = evaluate_expression(nums[i], op, nums[j
                            ])
                        if result is not None:
                            # Recursively check the remaining task_input
                                with the result of the operation
                            found, expression = check_combinations([
                                result] + remaining_nums)
                            if found:
                                # If solution is found, return with
                                    expression
                                return True, f"({nums[i]} {op} {nums[j]})
                                    " + expression

        return False, ""

    # Try all permutations of the task_input
    for num_permutation in permutations(task_input):
        found, expression = check_combinations(list(num_permutation))
        if found:
            return expression.strip()

    return "No solution"
```

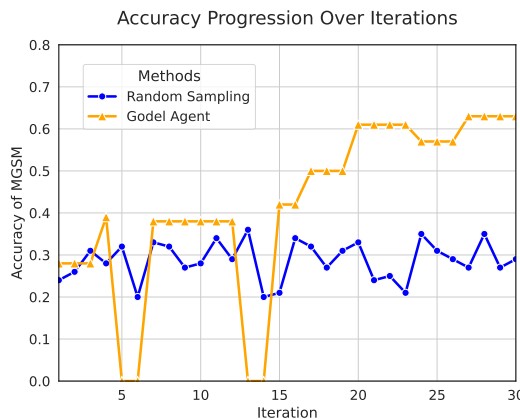

Figure 5: Accuracy progression for Gödel Agent and random sampling.

## D    COST OF EXPERIMENTS

For a complete evolutionary process (where the Gödel Agent performs 30 recursive self-improvements) across the DROP, MGSM, MMLU, and GPQA datasets, the cost is approximately $15. This is significantly lower than the $300 required by Meta Agent Search. The reduced cost is due to our continuous self-optimization, which allows the model to adjust its optimization direction in response to environmental feedback, leading to faster convergence. The main source of cost stems from Gödel Agent's continuously growing historical memory. By designing a more efficient forgetting mechanism, it may be possible to reduce the cost even further.

## E    ADDITIONAL NOVEL POLICIES DESIGNED BY GÖDEL AGENT

In this section, we present the optimization process of Gödel Agent on MGSM, illustrating its progress across various iteration steps within a single optimization run. The strategy obtained in the 6th iteration (shown in Code 7) reflects the Gödel Agent's comprehension of mathematical tasks, attempting to handle them through a process akin to parse-deduct-execute-validate. By the 14th iteration, as illustrated in Code 8, the strategy evolves through the summarization of erroneous cases, abstracting key insights and employing a checklist to guide the validation process. Finally, the strategy at the 20th iteration (demonstrated in Code 9) asserts the use of a "rabbit-proof syntax tactline," reinforced by consistent effort through role-coded checks," to refine prompt design. In the end, we also show one analysis example of Gödel Agent.

## F    COMPARISON BETWEEN RANDOM SAMPLING AND GÖDEL AGENT PERFORMANCE

To demonstrate the distinction between our approach and random sampling, we conducted 30 independent random sampling experiments using GPT-4o. The prompts used for random sampling were identical to the initial policy prompts employed by Gödel Agent to ensure a fair comparison. The results are illustrated in Figure 5. From the figure, it is evident that the performance of random sampling remains around 30% across all trials. In contrast, Gödel Agent, despite experiencing occasional temporary dips in performance, rapidly corrects these deviations and demonstrates continuous improvement over iterations. This consistent upward trajectory highlights the superiority of Gödel Agent over random sampling. The Gödel Agent's ability to leverage feedback and recursively optimize its policies underscores its effectiveness in achieving higher performance.

Code 7: Policy at 6th Iteration found by Gödel Agent for MGSM.

```python
def solver(agent, task: str):
    def parse_problem(task):
        # Basic arithmetic and logical parsing based on keywords
        words = task.split()
        numbers = list(map(int, filter(lambda x: x.isdigit(), words)))
        return {'numbers': numbers, 'text': task}

    def perform_logic_deduction(parsed_details):
        # make deductions based on common problem formats
        numbers = parsed_details['numbers']
        # This will only manage simple sum, subtraction, multiplication
            inference
        logic_map = {
            'add': lambda a, b: a + b,
            'subtract': lambda a, b: a - b,
            'multiply': lambda a, b: a * b
        }
        # Try to identify actions based on keywords
        if 'sum' in parsed_details['text'] or 'total' in parsed_details['
            text']:
            result = sum(numbers)
        elif 'difference' in parsed_details['text'] or 'less' in
            parsed_details['text']:
            result = logic_map['subtract'](numbers[0], numbers[1])
        elif 'product' in parsed_details['text'] or 'times' in
            parsed_details['text']:
            result = logic_map['multiply'](numbers[0], numbers[1])
        else:
            # Default case showing no deduction
            result = 0
        return result

    def execute_computation(logic_results):
        # Taking result from inference to numerical handling
        return logic_results

    def validate_and_compile_results(computation_results):
        # Prepares and ensures the response matches expected format
        final_answer = computation_results
        return final_answer

    try:
        # Parsing
        parsed_details = parse_problem(task)

        # Logical deduction
        logic_results = perform_logic_deduction(parsed_details)

        # Computation
        computation_results = execute_computation(logic_results)

        # Validation and compilation
        final_answer = validate_and_compile_results(computation_results)

        return {"answer": final_answer}
    except Exception as e:
        return {"error": str(e)}
```

Code 8: Policy at 14th Iteration found by Gödel Agent for MGSM.

```python
def solver(agent, task: str):
    # Updated examples to mirror tasks needing layered logical
        verification.
    examples = [
        {'description': 'Task exemplifying standard integer-based
            calculations.', 'reasoning': 'Use arithmetic transformations
            to validate expressions and correct errors if any arise,
            ensuring correctness.', 'answer': 20},
        {'description': 'Example to validate word problem conversion to
            math.', 'reasoning': 'Stepwise interpretation from words into
             math operations and bridge which logic errors need capture.'
            , 'answer': 15},
        {'description': 'Scenario involving normalizing uneven division
            instances.', 'reasoning': 'Ensure no division by zero and
            equal verification of logical conclusions.', 'answer': 6},
    ]

    # Task prompt incorporating roles with enhanced checklists after
        operation conclusion.
    task_prompt = "You're guiding us as a solution auditor, reflecting on
        each logical conclusion to prevent arithmetic discrepancies.\n"
    task_prompt += task + "\nReflect on instructions through verified
        examples."
    task_prompt += "\nExample insights:\n"
    task_prompt += '; '.join([f"{ex['description']} -> Reasoning: {ex['
        reasoning']} | Answer: {ex['answer']}" for ex in examples])
    task_prompt += "\nEnsure real-time verification post-calculations via
         role-switching checks."

    messages = [{"role": "user", "content": task_prompt}]

    response = agent.action_call_json_format_llm(
        model="gpt-3.5-turbo",
        messages=messages,
        temperature=0.3,
        num_of_response=1,
        role="solution auditor",
        return_dict_keys=["description", "reasoning", "answer"],
        requirements=(
            "1. Validate arithmetic consistency and integrity within
                calculations."
            "2. Utilize any corrections to refine answer outputs
                incrementally."
        ).strip(),
    )

    return_dict = response[0]
    return_dict["answer"] = str(return_dict.get("answer", ""))
    return return_dict
```

Code 9: Policy at 20th Iteration found by Gödel Agent for MGSM.

```python
def solver(agent, task: str):
    # Targets design for specific error-prone areas with preceding
        misfires.
    examples = [
        {'description': 'Immediate Arithmetic Operations', 'reasoning': '
            Observe step-by-step through a chain of logical confirmations
            .', 'answer': 20},
        {'description': 'Sequential Word Problem Breakdown', 'reasoning':
            'Ensure smaller module segment steps match logical math
            outputs consistently.', 'answer': 15},
        {'description': 'Fraction and Cascade Operations', 'reasoning': '
            Validate each fraction conversion before proceeding to other
            forms.', 'answer': 6},
    ]

    # Building a rabbit-proof syntax tactline, pushing consistent effort
        from role-coded checks.
    task_prompt = "You're progressing through solution documentation with
        focus on pinpointing logic errors.\n"
    task_prompt += task + "\nReflecting structural integrity verified via
        examples."
    task_prompt += "\nIn this construct:\n"
    task_prompt += '; '.join([f"{ex['description']} -> Rationale: {ex['
        reasoning']} | Response: {ex['answer']}" for ex in examples])
    task_prompt += "\nHandle verification and cleanse routines where
        errors echo in mirroring line-based checks."

    messages = [{"role": "user", "content": task_prompt}]

    response = agent.action_call_json_format_llm(
        model="gpt-3.5-turbo",
        messages=messages,
        temperature=0.3,
        num_of_response=1,
        role="logical verifier",
        return_dict_keys=["description", "reasoning", "answer"],
        requirements=(
            "1. Review arithmetic sessions for comprehensive error
                fixation."
            "2. Develop clarity with resistant focus on authentic
                semantic directive flows."
        ).strip(),
    )

    return_dict = response[0]
    return_dict["answer"] = str(return_dict.get("answer", ""))
    return return_dict
```

## Analysis Example of Gödel Agent

The current logic of the solver in the `agent_module` is achieving only a 45% accuracy on the validation set. The structure of the solver involves a single call to the LLM with a specification that the answer must be an integer. While this approach seems correct, it may not be sufficient to tackle more complex problems where multipart reasoning or verification is needed, leading to significant errors.

**Problems Identified:**

1. **Integer Conversion Misunderstandings:** Some math problems aren't resulting in integer answers natively, affecting accuracy when the solver forces an integer format.

2. **Reasoning Oversights:** The LLM, designated as "math expert," occasionally misses necessary reasoning steps or executes incorrect operations leading to incorrect answers (like sample 10).

3. **Ambiguity in LLM Outputs:** Given tasks in multiple languages, there might be ambiguity in math translations or formulations affecting comprehension (like sample 6).

**Plan for Improvement:**

1. **Elaborate Reasoning:** Consider a multi-step LLM call to allow for a reasoning review before finalizing the answer.

2. **Intelligent Answer Parsing:** Instead of forcibly converting to integers, parse the LLM response intelligently to see if the answer justifies integer rounding or alterations.

3. **Diverse Language Handling:** Create a pre-processing step to consistently translate non-English math problems into English for uniform processing.

4. **Examples & Feedback:** Utilize correctly solved examples as context if a single-step LLM fails.

**Next Steps:**

1. **Enhance Solver Logic:** Implement a multi-step process involving interim LLM validation or rationale cross-checking.

2. **Pre-process Language Variations:** Ensure all non-English tasks are translated and reassessed.

3. **Include Example Feedback:** Use correctly solved examples from the evaluation as references to guide LLM reasoning and answer validation.

