# OpenReview forum: "Gödel Agent: A Self-Referential Framework Helps for Recursively Self-Improvement"
_ICLR.cc/2025/Conference — Submitted to ICLR 2025_

### Official Review · Reviewer_J1mU · 2024-10-28

**Soundness:** 2
**Presentation:** 2
**Contribution:** 2
**Rating:** 5
**Confidence:** 4

**Summary:**

The paper presents the Godel Agent, a prompt engineering framework inspired by the Godel machine, claiming to be a self-improvement framework. The authors evaluate Godel Agent in 4 tasks, and the game of 24.

**Strengths:**

The research topic is interesting and important. The paper provides an extensive evaluation.

**Weaknesses:**

1. There are several ambitious claims within the paper; for example, the abstract suggests that the method can explore the entire space to find the optimal solution. This is a bold statement and does not seem to be substantiated by the findings presented.
2. The central claim about the capability to create "different agents" from an initial agent mostly involves modifying the conditioning of the autoregressive order through prompt engineering.
3. Figure 1 shows some learnable components, yet the framework itself does not appear to incorporate learnable elements.
4. Line 50 claims "to eliminate the human design prior," which is challenging given that LLMs inherently contain human priors from being trained on human-generated text.
5. Discussions on self-awareness of LLMs are speculative and might be better left out of a scholarly paper focused on introducing a new method.
6. Section 3 is notably small and lacks clarity on the method’s implementation. It would benefit from additional details, particularly on how self-improvement is enacted within the Godel Agent and how evaluations are conducted.
7. Lines 215-233 imply that evaluations rely on the environments provided. If this involves using the test set for evaluation, it represents a significant limitation of the method.
8. How the solutions in Godel Agent are evaluated (referenced in line 6 of the pseudocode) needs clearer explanation.
9. It would be beneficial to compare the cost of the Godel Agent with other methods, such as the CoT, both in terms of time and financial resources.
The authors might want to reevaluate their claims and revise the paper to provide a clearer presentation of the proposed method.

**Questions:**

NA

---

> ### Author Response · Authors · 2024-11-15
> **Official Response to Reviewer  J1mU by Authors -- Part One**
>
> Thanks for your careful and valuable comments. We will explain your concerns point by point.
>
> > There are several ambitious claims within the paper; for example, the abstract suggests that the method can explore the entire space to find the optimal solution. This is a bold statement and does not seem to be substantiated by the findings presented.
> >
>
> In theory, Gödel Agent is designed to explore the solution space and find the optimal solution within the computational resources available, as demonstrated in the Gödel machine paper[1]. However, due to practical limitations, such as computational resource constraints and the inherent instability of LLM decision-making, it is possible that the method may not always achieve the theoretical optimal in real-world applications.
>
> We appreciate your critique, and we will revise the paper to express this point more clearly and cautiously. Specifically, we will emphasize that while Gödel Agent can theoretically explore and optimize solutions within the available resources, practical challenges may affect the extent to which this optimality is achieved in real-world scenarios.
>
> > The central claim about the capability to create "different agents" from an initial agent mostly involves modifying the conditioning of the autoregressive order through prompt engineering.
> >
>
> It seems there might be a misunderstanding regarding the core mechanism of Gödel Agent. To clarify, we do not create "different agents" through prompt engineering. Rather, there is a single Gödel Agent that continuously refines and improves itself over multiple iterations.
>
> The key difference is that Gödel Agent does not simply adjust its behavior through prompt modifications. Instead, it engages in a recursive process where the agent modifies its own underlying code to optimize its capabilities. For example, the initial agent $a_0$ uses its current state to improve itself, resulting in $a_1$. Then, $a_1$ uses its enhanced capabilities to further refine itself into $a_2$, and so on. This self-referential and recursive improvement process goes beyond trivial prompt adjustments—it fundamentally changes the agent’s optimization abilities over time.
>
> > Figure 1 shows some learnable components, yet the framework itself does not appear to incorporate learnable elements.
> >
>
> To clarify, Gödel Agent’s framework indeed incorporates learnable elements. The action space, tools, and policies are not static but are dynamically modifiable by the agent. Gödel Agent can autonomously refine its action space, create and adjust its tools, and optimize its policies based on feedback during its self-improvement process. These components are effectively learnable because they can be adapted and optimized over time as the agent evolves.
>
> > Line 50 claims "to eliminate the human design prior," which is challenging given that LLMs inherently contain human priors from being trained on human-generated text.
> >
>
> You are absolutely correct that LLMs inherently carry human priors because they are trained on human-generated text. However, in line 50, our intention was to convey that we aim to reduce the human priors specifically in the design of the agent’s logic, strategies, and decision-making processes. While LLMs themselves are inevitably influenced by their training data, our goal is to allow the agent to operate autonomously, minimizing manual human intervention in its policy design. In other words, while the underlying LLM contains human priors, the agent’s higher-level decision-making and optimization processes are designed to evolve independently, free from predefined human-designed rules or biases. We acknowledge that complete elimination of human influence is not feasible due to the nature of LLMs, but our focus is on reducing human biases in the agent’s self-improvement framework.
>
> > Discussions on self-awareness of LLMs are speculative and might be better left out of a scholarly paper focused on introducing a new method.
> >
>
> Thank you for your thoughtful comment and suggestion. We apologize for any confusion caused by our use of the term "self-awareness." In this context, we intended to convey that Gödel Agent has the capability to introspect and read its own code and files, not to imply any philosophical sense of consciousness or awareness. We have emphasized this point more clearly in the revised version of the paper. If you find that the current explanation is still not sufficiently clear, would using the term of *self-introspection* be more appropriate?

---

> ### Author Response · Authors · 2024-11-15
> **Official Response to Reviewer J1mU by Authors -- Part Two**
>
> > Section 3 is notably small and lacks clarity on the method’s implementation. It would benefit from additional details, particularly on how self-improvement is enacted within the Godel Agent and how evaluations are conducted.
>
> The core idea behind Gödel Agent is to create a self-referential system that can recursively modify its own components for continuous self-improvement. The initial setup focuses on enabling this recursive self-modification capability. Once the system is initialized, the LLM autonomously decides which modules to modify based on feedback from its current performance.
>
> Regarding the self-improvement process, the LLM monitors the rewards from its current policy as well as its internal state information. Based on this feedback, the LLM generates action instructions that can modify various components of the agent, such as its policies, tools, or even its decision-making logic. This recursive process allows Gödel Agent to iteratively optimize itself over time.
>
> As for evaluations, after Gödel Agent has optimized its policy, the resulting algorithm is tested on the target task’s test set. We use standard evaluation metrics such as accuracy or F1 score to assess the performance.
>
> > Lines 215-233 imply that evaluations rely on the environments provided. If this involves using the test set for evaluation, it represents a significant limitation of the method.
> >
>
> The agent does not have access to the test data. In fact, during the evolving process, we only utilize validation data, as detailed in Appendix B. To ensure a fair comparison with baseline methods, the selection of validation data follows the approach used in the paper[2]. We use the scores on the validation set as feedback from the environment to guide the Gödel Agent in optimizing its policies.
>
> > How the solutions in Godel Agent are evaluated (referenced in line 6 of the pseudocode) needs clearer explanation.
> >
>
> The evaluation of solutions in Gödel Agent is conducted using an environment-provided utility function. Specifically, after Gödel Agent optimizes its policy, the resulting policy (which represents a task-specific algorithm) is tested on the test set of the target task. The utility function assesses the policy’s effectiveness by calculating standard performance metrics such as accuracy or F1 score.
>
> > It would be beneficial to compare the cost of the Godel Agent with other methods, such as the CoT, both in terms of time and financial resources. The authors might want to reevaluate their claims and revise the paper to provide a clearer presentation of the proposed method.
> >
>
> Thank you for your suggestion.  Our computational cost can be broken down into two parts:
>
> 1. **Self-Improvement Phase**: Gödel Agent incurs additional computational costs during the self-improvement phase, as detailed in Appendix D. Compared to approaches like ADAS, our method significantly reduces these costs by optimizing the agent's capabilities autonomously. While CoT-based methods do not require this self-improvement phase, they rely heavily on human experts for design, which involves a substantial time and financial investment. One of the main motivations for Gödel Agent is to replace this manual effort with computational self-optimization, thereby reducing the need for human intervention.
> 2. **Testing Phase**: The testing cost varies across different methods. For instance, methods like CoT-SC and debate require multiple queries and thus have significantly higher computational costs compared to standard CoT. In contrast, Gödel Agent's testing costs generally fall between those of CoT and CoT-SC. Notably, in some cases, Gödel Agent evolves strategies that reduce or even eliminate the need for LLM queries, achieving much lower computational costs.
>
> [1] Goedel Machines: Self-Referential Universal Problem Solvers Making Provably Optimal Self-Improvements
>
> [2] Automated Design of Agentic Systems

---

> > ### Comment · Reviewer_J1mU · 2024-11-24
> >
> > Thanks for the response. I think some of my concerns have been addressed and I will raise my score from 3 to 5. However, there are many related works in this area recently, such as OPRO, AgentSquare, Aflow, GPTSwarm. I agree with reviewer ALKV that the evaluation setup and baselines still need some improvement.

---

> > > ### Author Response · Authors · 2024-11-24
> > >
> > > Thank you for your thoughtful feedback and for raising your score. We’re glad to have addressed some of your concerns.
> > >
> > > Regarding the baselines, we’d like to clarify that certain works you mentioned, such as AgentSquare and Aflow, were proposed only in the past two months and are concurrent with our work, which is why they were not included in the comparison.
> > >
> > > For OPRO, while it focuses on using LLMs to optimize prompts based on feedback, its approach is fundamentally similar to ADAS, which uses LLMs to optimize agent code. Given this similarity and to ensure a fair and up-to-date comparison, we chose the more advanced ADAS as a baseline.
> > >
> > > Our framework, however, differs significantly from prior works. Unlike frameworks constrained by fixed meta-learning algorithms, our approach avoids such limitations, offering greater flexibility and adaptability.
> > >
> > > We hope this explanation clarifies our choices and highlights the uniqueness of our contribution. Please let us know if you have further questions.

---

### Official Review · Reviewer_7jQx · 2024-11-01

**Soundness:** 2
**Presentation:** 3
**Contribution:** 2
**Rating:** 6
**Confidence:** 4

**Summary:**

The authors propose a self-evolving framework, Gödel Agent, inspired by the Gödel machine, to enable recursive self-improvements of agents, without relying on predefined routines or fixed optimization algorithms.
The authors conduct experiments to validate the proposed approach.

**Strengths:**

1) A framework
The authors propose a self-evolving framework, Gödel Agent, inspired by the Gödel machine, to enable recursive self-improvements of agents, without relying on predefined routines or fixed optimization algorithms.

2) Experiments
The authors conduct experiments to validate the proposed approach.

**Weaknesses:**

1. Why LLM-based agent is a valid approach? No LLM is perfect. All LLMs make mistakes. What is reliable information? What is ground truth? Then why should we rely on LLMs to make decisions? For software engineering, no LLM can achieve a pass@1 of 100%. To make it worse, the popular evaluation method with several tests, is not perfect, so the evaluation results are not fully reliable.

2.
How to make LLM-based agent a valid approach? LLMs may help generate plausible solutions. But there should be a mechanism to filter out or fix incorrect solutions.

The current submission does not provide such a mechanism.

How to improve an LLM-based agent?
It seems this boils down to waiting for stronger LLMs, which is likely out of the scope of the paper, though.

3.
There are papers discussing / criticizing the reasoning / planning capacity of LLMs, which is the foundation of LLM-based agent (including those based on the Gödel machine), e.g.

LLMs can’t plan, but can help planning in LLM-modulo frameworks, in ICML, 2024.

GSM-Symbolic: Understanding the Limitations of Mathematical Reasoning in Large Language Models. arXiv 2024

Without sound capabilities for reasoning / planning, how to build LLM-based agents, in particular, the Gödel Agent?

It appears that empirical improvements are not enough to answer the above questions.

**Questions:**

see weaknesses

---

> ### Author Response · Authors · 2024-11-15
> **Official Response to Reviewer xqBT by Authors**
>
> Thanks for your insightful comments. We are happy to discuss with you about your points.
>
> > 1.  Why LLM-based agent is a valid approach? No LLM is perfect. All LLMs make mistakes. What is reliable information? What is ground truth? Then why should we rely on LLMs to make decisions? For software engineering, no LLM can achieve a pass@1 of 100%. To make it worse, the popular evaluation method with several tests, is not perfect, so the evaluation results are not fully reliable.
> >
> > 1. How to make LLM-based agent a valid approach? LLMs may help generate plausible solutions. But there should be a mechanism to filter out or fix incorrect solutions.
> >
> > How to improve an LLM-based agent? It seems this boils down to waiting for stronger LLMs, which is likely out of the scope of the paper, though.
> >
> > 1. There are papers discussing / criticizing the reasoning / planning capacity of LLMs, which is the foundation of LLM-based agent (including those based on the Gödel machine), e.g. 1) LLMs can’t plan, but can help planning in LLM-modulo frameworks, in ICML, 2024. 2) GSM-Symbolic: Understanding the Limitations of Mathematical Reasoning in Large Language Models. arXiv 2024
> >
> > Without sound capabilities for reasoning / planning, how to build LLM-based agents, in particular, the Gödel Agent?
> >
> > It appears that empirical improvements are not enough to answer the above questions.
> >
>
> We fully acknowledge that LLMs have limitations, make mistakes, and have significant room for improvement. However, it is also undeniable that LLMs have achieved remarkable results across many tasks and continue to improve. In our work, we propose utilizing LLMs to enable Gödel Agent’s self-improvement, focusing on leveraging their strengths rather than dwelling on their imperfections.
>
> Gödel Agent’s approach doesn’t rely solely on LLMs for decision-making. Instead, it incorporates LLMs as part of a recursive, self-improvement process where the agent can autonomously refine its policies and improve its performance over time. This self-referential mechanism allows Gödel to evolve beyond the limitations of the initial LLM and address its shortcomings through continuous self-modification.
>
> We also want to emphasize that our framework is not tied exclusively to LLMs as decision tools. If alternative methods for decision-making exist, our framework is flexible enough to incorporate them, which makes it adaptable to different settings and tools.
>
> We ask that you focus on evaluating the innovative contributions of our framework and the novel approach it offers for agent self-improvement. While we are open to further discussion of LLMs' limitations, we kindly request that you reconsider the evaluation of our paper based on the broader contributions of our approach, as it provides a unique perspective on agent self-improvement. Thank you again for your feedback.

---

> > ### Comment · Reviewer_7jQx · 2024-11-24
> > **the proposed method does not provide a feasible mechanism to collect reliable signals**
> >
> > Model improvements require reliable signals. However, the proposed method does not provide a feasible mechanism to collect such signals.
> >
> > The capability of self-inspect is limited by the current AI, in particular, LLMs, which are not perfect.
> >
> > The interactions with the environment may be very costly for model improvements. (This appears similar to reinforcement learning.)
> >
> >
> > The proposed Godel Agent depends on the capacities of LLMs and program synthesis, both are in progress and far from perfect, and breakthroughs are called for.
> >
> > Neither LLMs nor program synthesis can guarantee reliable signals for the proposed method to make reliable improvements.
> >
> >
> > Line 153 says the dependence on an LLM
> >
> > Line 450
> > This suggests that, given the current limitations of LLMs, it is challenging for Go ̈del Agent to innovate beyond state-of-the- art algorithms. Improvements in LLM capabilities are anticipated to unlock more innovative self- optimization strategies in the future.

---

> > > ### Author Response · Authors · 2024-11-27
> > > **Kindly request for reconsideration and focus on our framework design**
> > >
> > > Dear Reviewer 7jQx,
> > >
> > > Thank you for your time and efforts in reviewing our paper and responses.
> > >
> > > After carefully considering your comments, we noticed that some points of critique appeared to focus predominantly on the use of LLMs rather than the framework design and its contributions. We kindly request you to reconsider your assessment with a focus on the originality and applicability of the proposed framework. Respectfully, we remain open to further discussions and wonder if you have any other concerns. We would be happy to address them promptly.
> > >
> > > Sincerely,
> > >
> > > Authors of Submission 13979

---

> ### Author Response · Authors · 2024-11-25
>
> - **Reliable Signals**: Gödel Agent uses performance on the validation set as a reliable optimization signal. This signal is quantitative, task-specific, and derived directly from the agent’s actions, ensuring relevance to the optimization process. Experimental results in Section 5 demonstrate that Gödel Agent can effectively improve its strategies based on this feedback.
> - **Cost of Interactions**:
> Gödel Agent minimizes the cost of interactions by requiring only one interaction per task. Once a policy is optimized, it can be reused for repeated task executions without additional interactions. As shown in Appendix D, the experimental costs are manageable and not prohibitive.
> - **Dependency on LLMs**:
> Gödel Agent’s reliance on LLMs reflects the current state of the field, where LLMs are central to many state-of-the-art AI systems. This dependency is not unique to Gödel Agent but is shared by a majority of cutting-edge methods.
>
> We believe Gödel Agent represents a novel and significant step forward, particularly in its ability to self-introspect, adapt, and optimize autonomously. We encourage the reviewer to consider these contributions and their potential implications. A fair and thoughtful assessment of the work’s value would be beneficial.
>
> If further clarification or additional evidence is needed, we are more than happy to provide it. We sincerely appreciate your effort in reviewing our work.

---

> ### Comment · Reviewer_7jQx · 2024-11-28
>
> From the Abstract: "Gödel Agent leverages LLMs to dynamically modify its own logic and behavior, guided solely by high-level objectives through prompting." My understanding is that LLMs are essential to Gödel Agent.
>
> I adjust the score a little bit, to appreciate the authors' hard work.
> However, I'm concerned with how the LLM community is approaching AI agents, as discussed earlier.

---

> > ### Author Response · Authors · 2024-11-28
> >
> > Gödel Agent, like other state-of-the-art agents, requires a foundational model to support its decision-making process. Given the strong performance of LLMs across many tasks, we, like other works [1,2,3,4,5,6,7], chose to use an LLM as the core model for Gödel Agent. We believe this reliance on LLMs should not be seen as a limitation but rather as leveraging the best available technology to achieve our goals.
> >
> > Respectfully, we hope you can reconsider your assessment.
> >
> > [1] Nova: An Iterative Planning and Search Approach to Enhance Novelty and Diversity of LLM Generated Ideas
> >
> > [2] Investigate-Consolidate-Exploit: A General Strategy for Inter-Task Agent Self-Evolution
> >
> > [3] Reflexion: Language Agents with Verbal Reinforcement Learning
> >
> > [4] Trial and Error: Exploration-Based Trajectory Optimization for LLM Agents
> >
> > [5] Voyager: An Open-Ended Embodied Agent with Large Language Models
> >
> > [6] AFlow: Automating Agentic Workflow Generation
> >
> > [7] Language Agent Tree Search Unifies Reasoning Acting and Planning in Language Models
> >
> > ......

---

> > ### Author Response · Authors · 2024-11-28
> >
> > Thank you for adjusting the score and for recognizing our efforts. We truly appreciate your thoughtful feedback throughout the review process.
> >
> > We understand your concerns about how the AI community and we agree that there is room for broader discussions about LLM and Agent.

---

> > > ### Comment · Reviewer_7jQx · 2024-11-29
> > > **Constructive suggestion**
> > >
> > > The following are a sample of papers which may help you reconsider the method and direction for LLM agents.
> > >
> > > Position: LLMs Can’t Plan, But Can Help Planning in LLM-Modulo Frameworks, ICML 2024
> > >
> > > On the Brittle Foundations of ReAct Prompting for Agentic Large Language Models, arXiv 2024
> > >
> > > Chain of Thoughtlessness? An Analysis of CoT in Planning
> > >
> > > Can Large Language Models Really Improve by Self-critiquing Their Own Plans?
> > >
> > > AI Agents That Matter, arXiv 2024
> > >
> > > When is Tree Search Useful for LLM Planning? It Depends on the Discriminator, ACL 2024
> > >
> > > TravelPlanner: A Benchmark for Real-World Planning with Language Agents, ICML 2024
> > > “the current language agents are not yet capable of handling such complex planning tasks-even GPT-4 only achieves a success rate of 0.6%”
> > >
> > > There are also lots of papers discussing the limitations of LLMs, e.g.,
> > >
> > > Faith and Fate: Limits of Transformers on Compositionality, NeurIPS 2023
> > >
> > > Embers of Autoregression: Understanding Large Language Models Through the Problem They are Trained to Solve, PNAS 2024
> > >
> > > Can Large Language Models Infer Causation from Correlation? ICLR 2024

---

> > > > ### Author Response · Authors · 2024-11-29
> > > >
> > > > Thank you for your thoughtful feedback and the references you provided. We agree that LLMs, as highlighted in the papers, face challenges in tasks such as planning, reasoning, and self-correction. However, we would like to clarify the key aspects of Gödel Agent that address these concerns.
> > > >
> > > > (1) Unlike standard LLM-based approaches, Gödel Agent leverages performance signals from an external environment for feedback. This validation process helps ensure that the signals guiding the agent’s self-improvement are reliable. The feedback loop significantly enhances the quality of its improvements.
> > > >
> > > > (2) Gödel Agent follows the "Generate-Test-Critique" framework discussed in *LLMs Can’t Plan, But Can Help Planning in LLM-Modulo Frameworks*. Rather than using LLMs to directly plan or solve tasks, Gödel Agent uses an LLM-based optimization module to iteratively refine itself. The self-referential nature of the optimization module allows Gödel Agent to modify its own strategies and tools, leading to higher levels of performance and adaptability.
> > > >
> > > > (3) Gödel Agent does not rely on LLMs to solve tasks directly. Instead, it focuses on optimizing the policies or strategies used to tackle tasks. This decoupling from direct task-solving allows Gödel Agent to sidestep many of the issues LLMs face, such as compositionality and reasoning failures.
> > > >
> > > > We hope this clarifies our approach, and we appreciate the opportunity to discuss these important aspects further.

---

> > > > > ### Comment · Reviewer_7jQx · 2024-11-29
> > > > > **Is there an external verifier?**
> > > > >
> > > > > My understanding of the "Generate-Test-Critique" LLM-Modulo framework is that there should be an external verifier for outputs from LMMs.
> > > > >
> > > > > If only with environment feedback, it is something like the traditional RL approaches, which may face efficiency issues, and may not be feasible/effective for general problems.

---

> > > > > > ### Author Response · Authors · 2024-11-30
> > > > > >
> > > > > > Thank you for your clarification. You are correct that the "Generate-Test-Critique" LLM-Modulo framework typically involves an external verifier for LLM outputs (normally an answer for the question). Specifically, the output of Gödel Agent is the generated policy. Therefore, we use the external verifier to evaluate the effectiveness of the generated policy rather than individual outputs. This is done by assessing performance on a validation set, which provides feedback to refine the overall strategy.
> > > > > >
> > > > > > We hope this clarifies the role of external verification in Gödel Agent’s framework.

---

> > > > > > > ### Comment · Reviewer_7jQx · 2024-11-30
> > > > > > >
> > > > > > > Thanks for the clarification.
> > > > > > > It make sense to some extent, although I do not think it is an "external verifier" in a strong sense.
> > > > > > > Considering also other reviews, I adjust the score a little bit.

---

> ### Author Response · Authors · 2024-11-30
>
> Thank you for adjusting the score! This work represents our first step in exploring the framework of self-referential agents, and we are committed to further advancing this direction in future research. We greatly appreciate your insights regarding reasoning and planning with LLMs, and we will carefully consider the role of LLMs in agent frameworks as we move forward. Thank you again for your valuable input.

---

### Official Review · Reviewer_xqBT · 2024-11-01

**Soundness:** 3
**Presentation:** 3
**Contribution:** 2
**Rating:** 6
**Confidence:** 5

**Summary:**

This paper proposes a new self-evolving framework, Godel Agent, which leverages LLMs to dynamically modify self-generated logic and behavior. Empirical experiments show that this framework enables agents to recursively improve themselves without predefined routines or fixed optimization algorithms.

**Strengths:**

1. The workflow proposed in this paper is interesting; the core idea of recursive improvement is not implemented by improving the response itself step-by-step, but through updating the policy in an indirect way.

2. The paper is well written, and easy to follow.

3. The illustrations in figure 1 and examples given in Appendix C are clear and informative.

**Weaknesses:**

1. The motivation for adding four additional tools is not well-justified; there is no evidence supporting that these are the correct tools to add or that they comprehensively cover all needs. It's counterintuitive because if we claim the model has the ability to refine its policy and develop new policies, why do we need to provide tool guidance? Additionally, it's notable in Appendix C that although there is variability in the best policy, this policy seems can be decomposed into a combination of the listed tools, which suggests this work is not fully distinguished from meta-learning optimized agents where humans manually define the high-level strategy, and the agent learn a strategy to combine or rank the high-level strategy.

2. The current framework heavily relies on the model's capability to generate and refine policies, as well as generate responses given a policy and query. However, this would not improve the model's ability to acquire new knowledge if such kind of off-line dataset available.

3. The current refinement requires an oracle to progress the performance (U in line 6 from Algorithm 1). For the current Table 1, is the number of oracle utility function calls the same across all methods, and is the number of model queries the same for different settings?

**Questions:**

1. Based on Appendix C, the best policy found by Godel Agent for MMLU is the same as CoT-SC, why are the two numbers reported in Table 1 different? Similarly for the best policy for GPQA, why is it different from CoT? If this is due to extended few-shot examples or specific instructions, then it seems more like a prompt engineering problem, suggesting we're not using optimal prompts.

2. The reason why the method is evaluated on MGSM instead GSM8K is not well-justified, for my understanding, multi-lingual is not a target of the framework. Moreover, all current benchmark, even regular CoT has high accuracy, this together with limitation, I am questioning the generalizability of the framework on harder tasks, like MATH (https://github.com/hendrycks/math), it is reported that MATH data is contanmintated in some training dataset, so it would be safer to evaluate on MATH500 (https://huggingface.co/datasets/qq8933/MATH500), and omni-MATH (https://omni-math.github.io/). Additionally, the current evaluation would benefit from other agentic tasks, including code generation tasks, like SWEBench (https://www.swebench.com/).

3. For the related work, some recent recursive self-improvement work is not discussed
[1] Havrilla, Alex, et al. "Glore: When, where, and how to improve llm reasoning via global and local refinements." arXiv preprint arXiv:2402.10963 (2024).
[2] Hosseini, Arian, et al. "V-star: Training verifiers for self-taught reasoners." arXiv preprint arXiv:2402.06457 (2024).
[3] Qu, Yuxiao, et al. "Recursive introspection: Teaching language model agents how to self-improve." arXiv preprint arXiv:2407.18219 (2024).
[4] Kumar, Aviral, et al. "Training language models to self-correct via reinforcement learning." arXiv preprint arXiv:2409.12917 (2024).

4. How to safeguard the current workflow if the model is allowed to modify the policy and interaction code directly, and the policy will be used to generate further responses. How can we prevent runtime memory issues and ensure that both the code implementation and the final generation are not harmful?

---

> ### Author Response · Authors · 2024-11-15
> **Official Response to Reviewer xqBT by Authors -- Part One**
>
> Thanks for your careful and valuable comments. We will explain your concerns point by point.
>
> > The motivation for adding four additional tools is not well-justified; there is no evidence supporting that these are the correct tools to add or that they comprehensively cover all needs. It's counterintuitive because if we claim the model has the ability to refine its policy and develop new policies, why do we need to provide tool guidance? Additionally, it's notable in Appendix C that although there is variability in the best policy, this policy seems can be decomposed into a combination of the listed tools, which suggests this work is not fully distinguished from meta-learning optimized agents where humans manually define the high-level policy, and the agent learn a strategy to combine or rank the high-level strategy.
> >
>
> The motivation for initially providing Gödel Agent with four tools (Chain-of-Thought, error handling, code execution, and API calls) stems from the current limitations of existing LLMs. While Gödel Agent is designed to autonomously refine its strategies and develop new tools, the capabilities of current LLMs are not yet sufficient to handle more complex tasks without additional support. The inclusion of these tools serves to enhance the agent’s initial performance and practicality, given these constraints.
>
> It’s important to note that these tools are not hard-coded limitations. Gödel Agent has the flexibility to modify, extend, or even replace these tools as it optimizes itself. The tools provided are simply starting points to bridge current limitations, allowing Gödel to function effectively in its current environment. Over time, Gödel can autonomously evolve beyond the initial toolset as its capabilities improve.
>
> Regarding the concern that our approach is similar to meta-learning agents where humans define high-level strategies, we would like to emphasize a key distinction: Gödel Agent is not limited to combining predefined strategies. For instance, in the "Game of 24" task (Section 5.3), Gödel Agent developed a completely new search-based algorithm that achieved 100% accuracy. This represents a qualitative leap beyond what Meta Agent Search can achieve, as the latter is constrained by predefined search spaces and prompt-based strategies.
>
> > The current framework heavily relies on the model's capability to generate and refine policies, as well as generate responses given a policy and query. However, this would not improve the model's ability to acquire new knowledge if such kind of off-line dataset available.
> >
>
> The performance of Gödel Agent is indeed dependent on the capabilities of the underlying LLM to recognize that modifying its own code and structure will directly impact its performance.
>
> However, Gödel Agent does have the potential to surpass the limitations of the base LLM. If the LLM is advanced enough to understand how its own parameters influence the agent’s overall performance, Gödel could theoretically go further by generating code to fine-tune or even replace the underlying LLM. This self-modification could enable the agent to evolve beyond the capabilities of the original LLM, effectively enhancing its performance beyond its initial constraints.
>
> > The current refinement requires an oracle to progress the performance (U in line 6 from Algorithm 1). For the current Table 1, is the number of oracle utility function calls the same across all methods, and is the number of model queries the same for different settings?
> >
>
> Regarding the use of the oracle utility function, it is true that our method relies on an environment-provided utility function for optimization. During this phase, Gödel Agent autonomously decides when to call the utility function to obtain rewards on the validation set, similar to how human experts adjust their methods based on validation feedback. However, once the policy optimization is complete, the final policy uses the utility function in the same way as other methods, ensuring consistency across approaches in the evaluation phase.
>
> Regarding the number of model queries, it is indeed different across various methods. This variation is inherent due to the nature of each approach. For instance, CoT only requires a single query, while methods like CoT-SC and debate require multiple queries to achieve their results. As the framework we propose focuses on self-improvement, our primary concern is evaluating the effectiveness of each method rather than strictly equalizing the number of queries.

---

> ### Author Response · Authors · 2024-11-15
> **Official Response to Reviewer xqBT by Authors -- Part Two**
>
> > Based on Appendix C, the best policy found by Godel Agent for MMLU is the same as CoT-SC, why are the two numbers reported in Table 1 different? Similarly for the best policy for GPQA, why is it different from CoT? If this is due to extended few-shot examples or specific instructions, then it seems more like a prompt engineering problem, suggesting we're not using optimal prompts.
> >
>
> You are correct that the best policies found by Gödel Agent for MMLU and GPQA are similar in methodology to CoT-SC and CoT, respectively. The differences in the reported performance in Table 1 indeed stem from Gödel Agent’s use of extended few-shot examples and specific instructions.
>
> However, we do not view this as a limitation. On the contrary, this highlights Gödel Agent’s capability to autonomously optimize prompts, which includes fine-tuning few-shot examples and refining specific instructions to achieve better performance. If a human were to manually adjust prompts to optimize results, it would be a time-consuming and labor-intensive process. Gödel Agent, on the other hand, integrates this prompt optimization seamlessly into its self-improvement process. Thus, Gödel Agent is not only optimizing policies from a methodological standpoint but also refining prompts in a way that enhances performance, as demonstrated by our experimental results.
>
> > The reason why the method is evaluated on MGSM instead GSM8K is not well-justified, for my understanding, multi-lingual is not a target of the framework. Moreover, all current benchmark, even regular CoT has high accuracy, this together with limitation, I am questioning the generalizability of the framework on harder tasks, like MATH, MATH500,omni-MATH and SWEBench.
> >
>
> From a theoretical perspective, our framework is task-agnostic—provided that the environment can provide feedback, Gödel Agent can adaptively improve itself, making it applicable to both controlled and more complex real-world tasks, including embodied environments. However, we fully agree that experiments on more complex, real-world tasks would provide stronger evidence of the framework’s scalability. However, we chose the current set of tasks for the following reasons:
>
> 1. The baseline methods we are comparing against have only been tested on these controlled tasks, so using the same tasks ensures a fair and consistent comparison.
> 2. As this is the first proposal of such a framework, we wanted to demonstrate its feasibility and core capabilities on representative tasks before extending it to more complex scenarios. More complex tasks might distract from highlighting the novel aspects of our framework at this stage as they need more engineering work.
>
> > For the related work, some recent recursive self-improvement work is not discussed.
> >
>
> Thank you for pointing out these recent works. Below, we compare these approaches to our Gödel Agent and we have added them in our current version:
>
> 1. Glore proposes SORMs to improve LLM reasoning through global and local refinements, using synthetic data to predict correctness of solutions. It focuses on identifying when and where to refine reasoning steps but does not involve modifying the model's architecture or self-improvement mechanisms like Gödel Agent.
> 2. V-star trains a verifier to evaluate both correct and incorrect self-generated solutions, iteratively improving reasoning accuracy. Unlike Gödel Agent, which autonomously modifies its own architecture, V-star uses external verifiers to guide the refinement of reasoning.
> 3. RISE enables recursive self-improvement by fine-tuning models to introspect and correct previous mistakes in multiple iterations. Gödel Agent differs by incorporating a recursive mechanism that includes altering the architecture, not just refining responses.
> 4. SCoRe uses reinforcement learning to improve self-correction in LLMs by learning from self-generated correction traces, avoiding issues like behavior collapse.
>
> > How to safeguard the current workflow if the model is allowed to modify the policy and interaction code directly, and the policy will be used to generate further responses. How can we prevent runtime memory issues and ensure that both the code implementation and the final generation are not harmful?
> >
>
> Ensuring the safety of Gödel Agent during self-modification is indeed a top priority. To prevent potential misuse of resources, we recommend running Gödel Agent within a sandboxed environment where system permissions and resource access are tightly controlled. We have added an Ethics Statement at the end of the paper. Thank you for your suggestion!
>
> We hope that our responses have addressed your questions and concerns. We also hope that you find our approach exciting, as it represents a fundamental departure from previous methods. If you have any further questions, please feel free to reach out, and we will be happy to address them. Thank you once again for your valuable feedback.

---

> ### Comment · Reviewer_xqBT · 2024-11-16
> **Official Comment**
>
> > It's important to note that these tools are not hard-coded limitations. The Gödel Agent has the flexibility to modify, extend, or even replace these tools as it optimizes itself.
>
> While this potential flexibility is indeed intriguing, the current experimental results do not fully demonstrate this capability. With the exception of the "Game of 24" (which I will address separately), the observed behaviors primarily consist of combinations of the predefined tools detailed in Appendix C, rather than novel extensions or modifications.
>
> Regarding the "Game of 24" example, I have concerns about the claim that it "develops new strategies" beyond the predefined tools. Upon investigation, I found striking similarities between the generated algorithm and existing solutions. Specifically, when inputting the same problem from LeetCode (https://leetcode.com/problems/24-game/description/) into ChatGPT, the returned solution (https://chatgpt.com/share/67390c11-c9d0-8009-bd2d-ee221cc372fc) is remarkably similar, sharing even specific implementation details such as the 1e-6 threshold. This suggests that the model may be leveraging pre-trained knowledge rather than developing novel strategies through the proposed mechanism.
>
> These observations lead me to question whether the results presented in the paper fully support the authors' claims about the system's capabilities.
>
> > If the LLM is advanced enough to understand how its own parameters influence the agent's overall performance, Gödel could theoretically go further by generating code to fine-tune or even replace the underlying LLM.
>
> This theoretical possibility faces significant practical challenges. Given that even human experts currently struggle to fully understand how model parameters influence agent performance, it may not be prudent to base the method's validity on capabilities that require such advanced developments.
>
> > Regarding the use of the oracle utility function, it is true that our method relies on an environment-provided utility function for optimization.
>
> There appears to be an important distinction between two different applications of the oracle utility function that warrants clarification. While both the final policy generated by the Gödel agent and alternative strategies require oracle utility function calls during evaluation, the Gödel approach uniquely requires multiple calls during the policy development phase itself. This additional requirement during development could present significant practical limitations for general task applications, especially when compared to approaches like Chain of Thought (CoT) that do not need oracle utility function calls during their development phase.
>
>
> > The differences in the reported performance in Table 1 indeed stem from Gödel Agent's use of extended few-shot examples and specific instructions.
>
> >The baseline methods we are comparing against have only been tested on these controlled tasks, so using the same tasks ensures a fair and consistent comparison.
>
> I would like to seek clarification on two aspects of the experimental setup and baseline comparisons:
>
> Regarding dataset selection: The baseline methods cited in this paper (COT-SC, Self-Refine, LLM debate, Step-back-Abs) were actually evaluated on datasets GSM8K instead of MGSM. Could the authors explain the rationale behind this dataset selection?
>
> Concerning baseline performance: I notice that the baseline performance metrics reported in Table 1 are lower than those published in their respective original papers. This brings me to question about prompt optimization - could the authors clarify whether they used the optimal prompts from the original papers for these baseline implementations?
>
> The clarification on related work and safeguards makes sense to me, thanks!

---

> > ### Author Response · Authors · 2024-11-17
> > **Response to Official Comment - Part One**
> >
> > Thank you for your prompt response and we are happy to continue discussing our work with you.
> >
> > > While this potential flexibility is indeed intriguing, the current experimental results do not fully demonstrate this capability. The observed behaviors primarily consist of combinations of the predefined tools. Regarding the Game of 24, I have concerns about the claim that it "develops new strategies" beyond the predefined tools. The model may be leveraging pre-trained knowledge.
> >
> > We appreciate your recognition of the potential flexibility in Gödel Agent’s design. It is true that, based on our observations, Gödel Agent has not yet consistently discovered highly innovative strategies that go beyond human-designed policies. However, we would like to emphasize two important aspects:
> >
> > (1) While Gödel Agent does leverage existing tools and pre-trained knowledge to automate the design of policies, this alone represents a significant step forward. The ability to autonomously refine and combine these elements reduces the burden of manual design. Even if these strategies do not appear entirely novel, the process of automating policy refinement is still a meaningful advance.
> >
> > (2) We hope that the novelty of the Gödel Agent is evident. It is, to our knowledge, the first fully self-referential agent framework capable of modifying its own components without being constrained by predefined rules. The current limitations in observed innovation are largely influenced by the capabilities of existing LLMs. As these models become more powerful, we believe its potential for discovering novel strategies will increase significantly.
> >
> > > This theoretical possibility faces practical challenges. Given that even experts struggle to fully understand how model parameters influence agent performance, it may not be prudent to base the method's validity on capabilities that require such advanced developments.
> >
> > We agree that it is a challenging task and we do not expect current LLMs can achieve this level of optimization yet. The validity of our framework, however, does not rely on this theoretical capability. As shown by our current results, Gödel Agent is already effective at refining policies autonomously using the existing capabilities of LLMs. The idea of self-adjusting model parameters is presented as a potential future direction, highlighting the framework’s long-term potential rather than being a prerequisite for its current effectiveness.
> >
> > > There appears to be an important distinction between two different applications of the oracle utility function that warrants clarification. This additional requirement during development could present significant practical limitations for general task applications, especially when compared to approaches like CoT that do not need utility function calls during their development phase.
> >
> > We would like to clarify that methods like CoT and Debate are designed by experts. During their development, experts also rely on utility functions to test and adjust their strategies. The key difference is that Gödel Agent automates this process of strategy refinement, effectively replacing the human effort required with its own self-optimization using the utility function.
> >
> > We acknowledge that once CoTs are fully designed, they can be deployed without additional development costs. However, the same is true for the strategies generated by Gödel Agent. After the initial self-optimization phase, policies are fixed and can be applied to downstream tasks without further computational cost—much like CoT. Regarding whether Gödel Agent can generate a highly generalizable strategy similar to CoT, we believe this is an area that requires further exploration. As LLM capabilities continue to improve, we are optimistic that Gödel Agent will be able to produce even more generalized and robust strategies in the future.

---

> > ### Author Response · Authors · 2024-11-17
> > **Response to Official Comment - Part Two**
> >
> > > Regarding dataset selection: The baseline methods cited in this paper were actually evaluated on datasets GSM8K instead of MGSM. Could the authors explain the rationale behind this dataset selection?
> > >
> > > Concerning baseline performance: I notice that the baseline performance metrics reported in Table 1 are lower than those published in their respective original papers. This brings me to question about prompt optimization?
> >
> > Thank you for your questions and the opportunity to clarify our choices.
> >
> > Regarding the dataset selection: We followed the experimental setup used in paper [1], which includes both the choice of dataset (MGSM) and the evaluation of baseline methods. This was done to maintain consistency and ensure a fair comparison with previous work.
> >
> > As for the baseline performance metrics: We used the classic prompt designs, such as:
> >
> > - CoT: "Please think step by step and then solve the task."
> > - Self-Refine: "Please review the answer above and criticize where it might be wrong. If you are absolutely sure it is correct, output 'True' in 'correct'."
> > - Debate: "Given solutions to the problem from other agents, consider their opinions as additional advice. Please think carefully and provide an updated answer."
> > - Step-back: "What are the physics, chemistry, or biology principles and concepts involved in solving this task? First think step by step. Then list all involved principles and explain them."
> >
> > We did not heavily optimize these prompts. The differences in baseline performance compared to the original papers could be attributed to variations in prompt effectiveness when applied to different datasets. This variability highlights one of the key advantages of our framework: unlike fixed prompt-based methods, our approach can automatically adjust and optimize itself for different tasks.
> >
> > Furthermore, our results are consistent with those reported in paper [1], and our method significantly outperforms the baselines. Therefore, we believe these differences do not undermine the overall validity of our findings.
> >
> > Thank you once again for your interest in our work. Please feel free to reach out if you have any further questions
> >
> > [1] Automated Design of Agentic Systems https://arxiv.org/abs/2408.08435

---

> > > ### Comment · Reviewer_xqBT · 2024-11-17
> > > **Official Comment**
> > >
> > > Thank you for the detailed response. The clarification regarding dataset selection is helpful, but I maintain some concerns about the fundamental claims of the paper. Based on the presented results, the Gödel Agent appears to still operate within the constraints of predefined rules. The only instance where it generates a strategy that deviates from these predefined rules is in the "Game of 24" case. However, as discussed in my previous response, this particular case may be attributable to data contamination rather than novel strategy generation. Given that no other examples of genuine rule modification or extension have been presented, I find it difficult to conclude that the proposed method's novelty and effectiveness have been definitively demonstrated.
> > >
> > > Therefore, while I appreciate the potential of this direction, I believe my original assessment and score remain appropriate based on the current evidence presented in the paper.

---

> > > > ### Author Response · Authors · 2024-11-18
> > > >
> > > > Thank you for your thoughtful feedback. We understand that the examples presented in the main text may have given the impression that Gödel Agent operates within predefined rules. However, we would like to clarify that Gödel Agent is not constrained to such rules, and in fact, it demonstrates significant analytical and adaptive behavior during its optimization process.
> > > >
> > > > To address your concern, we have included several examples in Appendix E, showcasing how Gödel Agent refines its strategies across different iterations on the MGSM dataset (Note that these examples are selected randomly.). For instance:
> > > >
> > > > - **Iteration 6**: Gödel Agent developed a policy (Code7) that reflects its understanding of mathematical tasks by adopting a **parse-deduct-execute-validate** workflow to handle complex problems.
> > > > - **Iteration 14**: The policy evolved (Code8) to include insights derived from analyzing incorrect cases, leading to the abstraction of key observations and the use of a **checklist** to guide the validation process.
> > > > - **Iteration 20**: Gödel Agent refined its approach (Code9) with a novel prompt design, employing what it described as "Building a rabbit-proof syntax tactline" and leveraging role-coded checks for consistency.
> > > >
> > > > These examples illustrate that Gödel Agent is capable of continuously adjusting its strategies, exploring new directions, and iterating on its previous strategies based on feedback. This adaptive process shows that Gödel Agent is not limited by predefined rules but instead engages in a degree of exploration and self-directed improvement.
> > > >
> > > > While some of these exploratory strategies may not always be adopted in the final optimized solution, we believe that this ability to explore and adjust strategies is a meaningful demonstration of Gödel Agent’s potential for innovation. We hope these examples clarify that our method does indeed enable rule modification and extension beyond static, human-designed templates.

---

> > > > > ### Comment · Reviewer_xqBT · 2024-11-18
> > > > > **Official Comment**
> > > > >
> > > > > Appreciate the added examples, I have a couple of follow-up questions/comments:
> > > > > 1. Could the authors clarify the examples generated by the policy at iterations 14 (Code 8) and 20 (Code 9)? While it appears the model is attempting to generate few-shot examples, the current presentation of descriptions, reasoning, and answers seems difficult to interpret due to their apparently random nature.
> > > > > 2. It would be beneficial to see empirical evidence of whether the model is "continuously adjusting" or "continuously improving" its strategies. Specifically, I recommend including a performance curve showing iteration number on the x-axis and performance metrics using the policy generated at each iteration of running Godel Agent on the y-axis.
> > > > > 3. The Godel Agent framework bears strong similarities to self-refinement approaches, with the key distinction being the introduction of an intermediate policy generation step. Specifically, while classic self-refinement updates responses directly with feedback, Godel Agent updates policies. However, as noted in [1], models are unlikely to achieve continuous improvement through natural language feedback alone without additional training. The examples in Appendix E and Code 2 appear to show policy variation rather than systematic refinement. To validate the effectiveness of the sequential approach (curve in 2), I am also curious about comparing against a baseline where multiple policies are sampled simultaneously given the problem set, each policy is evaluated independently, and results are compared against the sequential feedback approach.
> > > > > 4. To support my argument 3, other than [1], recent work demonstrating successful sequential improvement has required fine-tuning [2, 3] to enable such capabilities in off-the-shelf models. This suggests two potential paths to enable effective "self-referential framework": training models specifically to learn self-improvement capabilities or implementing direct feedback mechanisms that allow back-propagation to the network.
> > > > >
> > > > > [1] Huang, Jie, et al. "Large language models cannot self-correct reasoning yet." arXiv preprint arXiv:2310.01798 (2023).
> > > > > [2] Qu, Yuxiao, et al. "Recursive introspection: Teaching language model agents how to self-improve." arXiv preprint arXiv:2407.18219 (2024).
> > > > > [3] Kumar, Aviral, et al. "Training language models to self-correct via reinforcement learning." arXiv preprint arXiv:2409.12917 (2024).

---

> > > > > > ### Author Response · Authors · 2024-11-20
> > > > > >
> > > > > > > The Godel Agent framework bears strong similarities to self-refinement approaches, with the key distinction being the introduction of an intermediate policy generation step. Specifically, while classic self-refinement updates responses directly with feedback, Godel Agent updates policies.
> > > > > >
> > > > > > It is important to emphasize that our approach is fundamentally different from classic self-refinement approaches. While self-refinement typically focuses on directly updating responses based on feedback, Gödel Agent operates at a higher level. The key distinction lies in its ability to modify not only the policies themselves but also the framework responsible for generating these policies—Gödel Agent itself.
> > > > > >
> > > > > > Specifically, Gödel Agent enhances its ability to refine policies by updating its own components, such as creating tools like assistant agents or numerical parsers, sampling multiple policies for evaluation, enhancing error feedback mechanisms and improving error-handling capabilities. This self-referential ability to improve both policies and the mechanisms that create them is the core innovation of Gödel Agent.
> > > > > >
> > > > > > > However, as noted in [1], models are unlikely to achieve continuous improvement through natural language feedback alone without additional training.
> > > > > > >
> > > > > >
> > > > > > Paper [1] proposes that LLMs struggle to self-correct their responses without external feedback. However, Gödel Agent uses the results from the validation set as direct feedback. For example, sometimes it performs detailed case analysis to identify discrepancies and understand the root causes of errors by comparing its outputs with ground truth answers. This systematic approach allows the agent to refine its policies effectively.
> > > > > >
> > > > > > > The examples in Appendix E and Code 2 appear to show policy variation rather than systematic refinement. To validate the effectiveness of the sequential approach (curve in 2), I am also curious about comparing against a baseline where multiple policies are sampled simultaneously given the problem set, each policy is evaluated independently, and results are compared against the sequential feedback approach.
> > > > > > >
> > > > > >
> > > > > > Thank you for this thoughtful suggestion. Based on our observations, Gödel Agent dynamically adjusts its approach to policy improvement. In most cases, it refines policies sequentially, but when refinement proves ineffective, it explores entirely new policies.
> > > > > >
> > > > > > To address your recommendation, we conducted a baseline comparison on the MGSM dataset. We sampled five policies generated by the LLM and tested them independently without sequential feedback. The results are shown below:
> > > > > >
> > > > > > | Policy 1 | Policy 2 | Policy 3 | Policy 4 | Policy 5 |
> > > > > > | --- | --- | --- | --- | --- |
> > > > > > | 29.7 | 25.4 | 31.3 | 28.9 | 32.6 |
> > > > > >
> > > > > > As seen, the performance of these sampled policies is comparable to baseline results reported in the paper and significantly lower than Gödel Agent’s sequential improvement score of 64.2. These results validate the advantage of Gödel Agent’s sequential feedback approach over static sampling of multiple policies.
> > > > > >
> > > > > > > To support my argument 3, other than [1], recent work demonstrating successful sequential improvement has required fine-tuning [2, 3] to enable such capabilities in off-the-shelf models. This suggests two potential paths to enable effective "self-referential framework": training models specifically to learn self-improvement capabilities or implementing direct feedback mechanisms that allow back-propagation to the network.
> > > > > > >
> > > > > >
> > > > > > We agree that fine-tuning and implementing feedback mechanisms for backpropagation are effective methods for improving model capabilities, as demonstrated in [2, 3]. However, we would like to clarify that these approaches do not constitute a true "self-referential framework" as defined in our work. In such approaches, the training code and feedback mechanisms themselves remain fixed and unmodifiable. What is updated are the outputs produced by these fixed components.
> > > > > >
> > > > > > In contrast, Gödel Agent’s framework allows the agent to autonomously modify not only the policies it generates but also the training code and feedback mechanisms responsible for optimizing these policies. Gödel Agent can inspect and rewrite the code used for its own improvement, including the mechanisms that provide feedback, creating a fully self-referential system. This capability goes beyond simply applying pre-defined optimization routines.
> > > > > >
> > > > > > While fine-tuning and direct feedback mechanisms could be incorporated into Gödel Agent as human priors to initialize its optimization module, we deliberately chose not to explore these methods in this work. As this is the first self-referential agent framework, we aimed to highlight Gödel Agent’s unique ability to autonomously create and modify all components of its system without relying on fixed external routines. Future work could explore the integration of these approaches within the Gödel framework to further enhance its capabilities.

---

> > > > > > > ### Comment · Reviewer_xqBT · 2024-11-20
> > > > > > > **Official Comment**
> > > > > > >
> > > > > > > Any response on:
> > > > > > > 1. What's the performance of the policy (Code7) on MGSM?
> > > > > > > 2. My previous Q1 "Could the authors clarify the examples generated by the policy at iterations 14 (Code 8) and 20 (Code 9)? While it appears the model is attempting to generate few-shot examples, the current presentation of descriptions, reasoning, and answers seems difficult to interpret due to their apparently random nature."
> > > > > > > 3. My previous Q2 "It would be beneficial to see empirical evidence of whether the model is "continuously adjusting" or "continuously improving" its strategies. Specifically, I recommend including a performance curve showing iteration number on the x-axis and performance metrics using the policy generated at each iteration of running Godel Agent on the y-axis."

---

> > > > > > > > ### Author Response · Authors · 2024-11-20
> > > > > > > >
> > > > > > > > Apologies for the confusion. Due to the length of the response in the draft, we mistakenly assumed it was part of the previous content and inadvertently deleted this section when submitting. Below are our responses to the three questions:
> > > > > > > >
> > > > > > > > > What's the performance of the policy (Code7) on MGSM?
> > > > > > > >
> > > > > > > > The performance of Code7 on MGSM is close to zero. This result is expected because Code7 relies solely on predefined rules to process mathematical problems and does not leverage any LLM capabilities. The purpose of including Code7 in our experiments was not to achieve strong results but to showcase Gödel Agent’s ability to explore beyond predefined human-designed rules.
> > > > > > > >
> > > > > > > > > Could the authors clarify the examples generated by the policy at iterations 14 (Code 8) and 20 (Code 9)? While it appears the model is attempting to generate few-shot examples, the current presentation of descriptions, reasoning, and answers seems difficult to interpret due to their apparently random nature.
> > > > > > > > >
> > > > > > > >
> > > > > > > > Based on our observations, the examples generated by Gödel Agent at iterations 14 (Code 8) and 20 (Code 9) are part of its validation process. The purpose of these examples is to assist the agent in reasoning and testing potential solutions. Specifically, the examples include three components: (1) A description of the validation purpose. (2) A detailed reasoning process or plan for the validation. (3) A final answer after completing the validation.
> > > > > > > >
> > > > > > > > In these iterations, the agent focuses on identifying common points in mathematical problems that require verification, such as specific calculations or logical steps. While the generated examples illustrate Gödel Agent's exploratory approach, we acknowledge that their quality may appear random. This is because these are randomly sampled outputs from a single cycle and were included to showcase the agent’s exploration of alternative strategies, rather than polished, final examples.
> > > > > > > >
> > > > > > > > > It would be beneficial to see empirical evidence of whether the model is "continuously adjusting" or "continuously improving" its strategies. Specifically, I recommend including a performance curve showing iteration number on the x-axis and performance metrics using the policy generated at each iteration of running Godel Agent on the y-axis.
> > > > > > > > >
> > > > > > > >
> > > > > > > > Thank you for the suggestion. As shown in Figure 4 of the paper, we have included a performance-iteration curve that illustrates the relationship between the iteration number (x-axis) and performance metrics (y-axis) for Gödel Agent. This figure also highlights key characteristics of the strategies generated at each iteration. We hope it will be helpful to you.
> > > > > > > >
> > > > > > > > We hope that our response will address your question. If you still have other questions, feel free to ask and we'll do our best to address your concerns.

---

> > > > > > > > > ### Comment · Reviewer_xqBT · 2024-11-24
> > > > > > > > > **Official Comment**
> > > > > > > > >
> > > > > > > > > I apologize for the delayed response. I understand that the rebuttal window will close shortly, I appreciate the authors' efforts to address the feedback and will carefully reevaluate the paper considering all comments provided during this rebuttal period.
> > > > > > > > >
> > > > > > > > > Regarding my main concern: If I understand correctly, the Godel Agent's Policy (Code 7) achieves near-zero performance at the 6th iteration for MGSM (from the last authors' response), while the random sampling approach achieves 32.6% performance at the 5th iteration (from the second last authors' response). Furthermore, none of the random sampling performances in the first five iterations fall below 25.4% (at least not near-zero). In other words, at least, given the current comparsion, Godel Agent is less effective than random sampling.
> > > > > > > > >
> > > > > > > > > Therefore, in an earlier response, I suggest including two comparative curves for MGSM: one showing the performance of each policy generated by the Godel Agent, and another showing the performance of randomly sampled policies. While this comparison may not be strictly fair due to Godel Agent's dynamic input length requiring larger FLOPs per sample, it would provide valuable insights into:
> > > > > > > > > 1. Whether the Godel Agent consistently improves its policy with increasing iterations?
> > > > > > > > > 2. Whether the higher performance is attributable to random chance, where increased sampling naturally leads to discovering better policies?
> > > > > > > > >
> > > > > > > > > Let me know if anything is unclear or I missed points in the paper.

---

> > > > > > > > > > ### Author Response · Authors · 2024-11-25
> > > > > > > > > >
> > > > > > > > > > Thank you for your reply. As shown in Figure 4 of the paper, Gödel Agent sometimes experiences temporary dips in performance, including near-zero results during exploratory attempts in certain iterations. This is part of the agent's design, as it autonomously explores unconventional strategies. However, Gödel Agent quickly corrects these deviations based on feedback, ensuring that the results recover in subsequent iterations. These temporary drops do not indicate that Gödel Agent is less effective than random sampling.
> > > > > > > > > >
> > > > > > > > > > In fact, Gödel Agent achieves a final performance of 64.2% on MGSM, which is significantly higher than the ~30% performance achieved by random sampling. This substantial improvement is far beyond what can be attributed to random chance or increased sampling alone. To address your suggestion, we have added comparative curves in Appendix F and Figure 5, showing the performance of Gödel Agent and random sampling across iterations on MGSM.
> > > > > > > > > >
> > > > > > > > > > Thank you for your valuable feedback, which has helped make the paper more complete. Please let us know if there are any additional points that require clarification.

---

> > > > > > > > > > > ### Comment · Reviewer_xqBT · 2024-11-25
> > > > > > > > > > > **Official Comment**
> > > > > > > > > > >
> > > > > > > > > > > Thanks for the clarification. The authors have addressed all my concerns, and I have increased my score accordingly. Good luck!

---

> > > > > > > > > > > > ### Author Response · Authors · 2024-11-25
> > > > > > > > > > > >
> > > > > > > > > > > > Thanks for appreciating our response and for updating the score. We greatly value your feedback and are happy to know that you found the response satisfactory.

---

### Official Review · Reviewer_waxH · 2024-11-04

**Soundness:** 3
**Presentation:** 3
**Contribution:** 3
**Rating:** 6
**Confidence:** 3

**Summary:**

* A big limitation of LLM agents is that there are handcrafted, self-improving agents are a promising direction to make this more autonomous
* self referential framework that enables agents to recursively improve themselves without predefined routines
* Agent can alter its own code and runtime memroy

The paper introduces Gödel Agent, a self-referential framework that enables agents to recursively improve themselves without predefined routines or fixed optimization algorithms.
Inspired by the Gödel machine, Gödel Agent allows agents to modify their own code and logic using large language models (LLMs), guided only by high-level objectives.
The agent uses techniques like monkey patching to read and alter its runtime memory, achieving self-awareness and self-modification.
Experiments across coding, science, and math tasks show that Gödel Agent outperforms manually crafted agents in performance, efficiency, and generalizability.

**Strengths:**

* Seems like a promising and open-ended approach for removing humans in the loop of agent pipeline building
* ADAS Meta Agent is a strong similar baseline and Godel outperforms it
* Demonstrated effective self-improvement across multiple domains including reasoning

**Weaknesses:**

* Experiments seem limited in scope—mostly on controlled tasks; might not scale to complex real-world applications or embodied tasks
* The final policies that are returned by the method don't seem very complicated or different from a basic human designed template. The agent designs discovered in the ADAS paper seem much more complex and creative. What explains this difference?
* Given the limited evaluations, the room for self-improvement seems limited, can we really get much better than the base model for mathematical reasoning with fancy agent pipelines and prompts? A better showcase of the method would be on openended or embodied tasks vs text based reasoning
* What’s the upper limit of self improvement, does it saturate?
   *Whats the improvement in the 6 iterations
* More detailed comparison to ADAS is needed in related work (Automated Design of Agentic Systems)
   * What is Meta Agent Search and why does this Godel agent perform better?
      * This needs to be clearer in the paper
* The method is somewhat vague and not clear what’s going on, and what this has to do with self reference/recursion. Is this different than just tasking an LLM with modifying its own agent code?

**Questions:**

How does Gödel Agent ensure safety and prevent harmful behaviors during self-modification? How do you put constraints on it such that it doesn't blow all of your GPT credits or hog all the GPUs on your system?

What mechanisms are in place to handle errors or prevent the agent from degrading its performance over time?

Can this approach scale to more complex tasks that require long-term planning or interaction with unpredictable environments?

How does the agent's performance depend on the underlying LLM's capabilities? Can it surpass those limitations?

---

> ### Author Response · Authors · 2024-11-15
> **Official Response to Reviewer waxH by Authors -- Part One**
>
> Thanks for your careful and valuable comments. We will explain your concerns point by point.
>
> > Experiments seem limited in scope—mostly on controlled tasks; might not scale to complex real-world applications or embodied tasks
>
> Thank you for your valuable feedback. From a theoretical perspective, our framework is task-agnostic—provided that the environment can provide feedback, Gödel Agent can adaptively improve itself, making it applicable to both controlled and more complex real-world tasks, including embodied environments. However, we fully agree that experiments on more complex, real-world tasks would provide stronger evidence of the framework’s scalability. However, we chose the current set of tasks for the following reasons:
> 1. The baseline methods we are comparing against have only been tested on these controlled tasks, so using the same tasks ensures a fair and consistent comparison.
> 2. As this is the first proposal of such a framework, we wanted to demonstrate its feasibility and core capabilities on representative tasks before extending it to more complex scenarios. More complex tasks might distract from highlighting the novel aspects of our framework at this stage as they need more engineering work.
>
> > The final policies that are returned by the method don't seem very complicated or different from a basic human designed template. The agent designs discovered in the ADAS paper seem much more complex and creative. What explains this difference?
>
> ADAS operates within a predefined search space, limited by human-designed methods, such as CoT and Debate. These algorithms are combined to create a solution, but the underlying structure is based on existing human knowledge. In contrast, Gödel Agent is not bound by such human priors. It is free to explore and design strategies that go beyond predefined templates, as demonstrated in the "Game of 24" task in Appendix C.2. In this task, Gödel Agent independently chose to use a search-based algorithm, breaking away from the limitations of prompt-based methods, as detailed in Section 5.3. While the policies Gödel Agent generates may not always seem highly complex in every instance, its full autonomy dictates that it has a higher ceiling and will get stronger as LLM grows.
>
> > Given the limited evaluations, the room for self-improvement seems limited, can we really get much better than the base model for mathematical reasoning with fancy agent pipelines and prompts? A better showcase of the method would be on openended or embodied tasks vs text based reasoning
>
> We agree that the full potential of Gödel Agent will be best demonstrated in more complex, real-world scenarios, such as open-ended or embodied tasks. We are excited about the possibility of applying our framework to these types of tasks in the future. Currently, our experiments have focused on text-based reasoning tasks to establish the feasibility and advanced capabilities of Gödel Agent in a controlled setting. While we recognize that this scope is limited, it was an essential first step in proving the underlying framework's effectiveness.
>
> > What’s the upper limit of self improvement, does it saturate? *Whats the improvement in the 6 iterations
>
> Thank you for your thoughtful question. The concept of recursive self-improvement does suggest that the process could continue indefinitely, as each iteration improves the optimizer, which in turn enhances the agent's ability to improve itself. However, in practice, we believe that there is an upper limit to this process. On the one hand, the core of Gödel Agent is based on a LLM, and the LLM’s understanding is inherently limited. If the agent becomes too complex, the LLM may no longer be able to comprehend or process its own changes, at which point the self-improvement process would stagnate. On the other hand, LLMs have some degree of randomness and are prone to errors, meaning that the optimization process may encounter hurdles that prevent indefinite improvement.
>
> Figure 4 in our paper shows an example of how the strategy changes as the number of iteration steps increases.

---

> ### Author Response · Authors · 2024-11-15
> **Official Response to Reviewer waxH by Authors -- Part Two**
>
> > More detailed comparison to ADAS is needed in related work (Automated Design of Agentic Systems) What is Meta Agent Search and why does this Godel agent perform better?
> >
>
> Thank you for the opportunity to clarify the comparison between our method and Meta Agent Search (MAS). In MAS, a fixed meta-agent is manually designed to create task-specific agents by interacting with the environment. However, this meta-agent remains static throughout the process, meaning the final optimized results are inherently constrained by the initial design and capabilities of the meta-agent.
>
> In contrast, our approach, inspired by the Gödel machine, moves beyond the meta-learning paradigm by endowing the agent with self-referential capabilities. This allows the agent to evolve not only its task-specific part but also its own optimization part. As a result, both the agent's ability to optimize and its performance on downstream tasks improve simultaneously, leading to more efficient and effective solutions.
>
> A clear example of this qualitative improvement can be seen in the "Game of 24" task as shown in Appendix C.2. Gödel Agent was able to design a search-based algorithm that solves the task with 100% accuracy, whereas MAS is limited to searching within predefined spaces and designing prompt-based methods. This shows that Gödel Agent can generate solutions that are not only more flexible but also fundamentally more capable than those produced by a fixed meta-agent.
>
> > The method is somewhat vague and not clear what’s going on, and what this has to do with self reference/recursion. Is this different than just tasking an LLM with modifying its own agent code?
> >
>
> In Gödel Agent, the concepts of self-reference and recursion refer to the agent's ability to autonomously modify its own code to improve both its optimization process and task-solving capabilities. The key difference lies in the fact that after each self-modification, Gödel Agent's optimization abilities themselves are enhanced. This creates a loop where improved optimization leads to more effective self-modification, which is recursively applied to the agent itself rather than an external entity.
>
> This process is fundamentally different from simply tasking an LLM with modifying its own agent code, as seen in methods like ADAS. In those cases, the LLM’s optimization capability remains static, meaning the changes are limited to task-specific adjustments without fundamentally enhancing the LLM's underlying optimization skills.
>
> > How does Gödel Agent ensure safety and prevent harmful behaviors during self-modification? How do you put constraints on it such that it doesn't blow all of your GPT credits or hog all the GPUs on your system?
> >
>
> Ensuring the safety of Gödel Agent during self-modification is indeed a top priority. To prevent potential misuse of resources, we recommend running Gödel Agent within a sandboxed environment where system permissions and resource access are tightly controlled. We have added an Ethics Statement at the end of the paper. Thank you for your suggestion!
>
> > What mechanisms are in place to handle errors or prevent the agent from degrading its performance over time?
> >
>
> Currently, Gödel Agent is equipped with an error feedback mechanism: when it performs an action that is invalid or non-functional, the system returns an error message. Gödel Agent uses this feedback to adjust its behavior and refine its strategies.
>
> Regarding performance degradation, we have chosen not to implement additional manual corrections. Instead, we rely on Gödel Agent’s ability to reflect on its actions and autonomously adjust its strategies over time. While it is technically possible to manually evaluate the strategy at each iteration and prevent modifications if performance declines, we have opted against this approach to minimize human intervention.
>
> Moreover, we believe that allowing the agent to occasionally make mistakes can be beneficial. By experiencing negative outcomes, Gödel Agent can learn which modifications are counterproductive, which can inform better decision-making in the future.

---

> ### Author Response · Authors · 2024-11-15
> **Official Response to Reviewer waxH by Authors -- Part Three**
>
> > Can this approach scale to more complex tasks that require long-term planning or interaction with unpredictable environments?
> >
>
> We believe that Gödel Agent’s approach can indeed scale to more complex tasks that require long-term planning and interaction with unpredictable environments. The key factor is the availability of feedback or rewards from the environment. As long as the environment can provide such feedback, Gödel Agent is capable of optimizing its strategies to adapt to the given task.
>
> > How does the agent's performance depend on the underlying LLM's capabilities? Can it surpass those limitations?
> >
>
> The performance of Gödel Agent is indeed dependent on the capabilities of the underlying LLM, particularly in terms of its decision-making, planning, and its ability to understand that it is part of Gödel Agent. The agent relies on the LLM to recognize that modifying its own code and structure will directly impact its performance.
>
> Regarding the second part of your question, Gödel Agent does have the potential to surpass the limitations of the base LLM. If the LLM is advanced enough to understand how its own parameters influence the agent’s overall performance, it could theoretically go further by generating code to fine-tune or even replace the underlying LLM. This self-modification could enable the agent to evolve beyond the capabilities of the original LLM, effectively enhancing its performance beyond its initial constraints.
>
> We hope that our responses have addressed your questions and concerns. If you have any further questions, please feel free to reach out, and we will be happy to address them. Thank you once again for your valuable feedback.

---

> > ### Comment · Reviewer_waxH · 2024-11-25
> > **Response to Official Responses**
> >
> > Thank you for your detailed responses addressing each of my concerns. While I appreciate the thorough explanations about Gödel Agent's theoretical capabilities and potential for complex tasks, I maintain my original score of 6. I still think the method is very promising! But the core limitation remains that all experiments are restricted to relatively simple text-based environments and mathematical reasoning tasks.
> > Although you argue the framework is task-agnostic, this needs to be demonstrated through concrete experiments in more challenging domains like embodied tasks or open-ended environments where the benefits of recursive self-improvement would be more apparent. Without such empirical validation, it's difficult to evaluate claims about the method's broader applicability and scalability.
> >
> > I look forward to seeing future work that tests these ideas in more complex settings.

---

> > > ### Author Response · Authors · 2024-11-26
> > >
> > > Thank you for your feedback and for recognizing the potential of Gödel Agent. We fully agree that more complex, real-world environments will better showcase the strengths of Gödel Agent’s recursive self-improvement capabilities. Transitioning to these more challenging tasks is primarily an engineering challenge, and we are committed to developing this aspect in future work.
> > >
> > > We would like to emphasize the innovative nature of Gödel Agent’s framework and its theoretical contributions. We hope you will reconsider the score, focusing on these contributions. Your feedback means a lot to us, and this reconsideration is crucial as we continue to develop this promising framework. We truly appreciate your thoughtful evaluation and are grateful for your ongoing interest in this line of research.
> > >
> > > Thank you again for your time and consideration.

---

### Official Review · Reviewer_ALKV · 2024-11-04

**Soundness:** 1
**Presentation:** 2
**Contribution:** 2
**Rating:** 5
**Confidence:** 3

**Summary:**

The paper proposes a self-improving language model agent ("Godel agent") and evaluates it on a variety of benchmarks. The results suggest that the self-improving agent achieves better performance than hand-designed agent scaffolds or meta-learned agents, while also being cheaper to run than the latter.

**Strengths:**

* The proposed method is conceptually simple and clear and provides and intriguing possibility of building self-improving agents with current LLM technology.
* The experiments include multiple benchmarks, a number of baselines and generally suggest strong results.
  * The authors run ablations of different components of their agent and find all of them improve performance.
* The qualitative analysis of the results is insightful and provides a good understanding of the strategies the self-improving agent implements.

**Weaknesses:**

**Results**
I'm concerned about the presentation of the experiments and the **fairness of the empirical evaluation**. My key concerns are that the Godel agent uses significantly more inference compute than any of the other methods and that the description of the evaluation protocol is not sufficient to determine if the evalaution is fair.

Here is my understanding of the evaluation setup (please correct me if I'm wrong):
* The Godel agent is run for N iterations on a validation set of samples. In each iteration is produces a new policy and a new learning function.
* After N iteration the policy is evaluated on held-out test problems and this result is reported.

This process gives the Godel agent significantly more inference compute than any of the hand-designed agent systems, which makes the comparison unfair. The only fair comparison in Table 1 is to Meta Agent Search which seems to typically be comparable in performance. Based on these results, I am not convinced the the Godel agent actually improves performance in a fair comparison.

This concern could be addressed by letting the baseline be a best-of-N method that applies N somewhat random perturbations to the policy and chooses the best of them. For example, this could be implemented by using GPT-4 to generate N different prompt variations and choose the best according to performance on the validation data.

Relatedly, according to Appendix B the Godel agent uses GPT-4 for the learning algorithm but all methods only use GPT-3.5 for the policy. This makes the comparison additionally skewed in favor of the Godel agent which has access to GPT-4 in contrast to the other methods. A fair comparison would use GPT-4 for both the learning algorithm and the policy.


**Presentation**

I also have some concerns about the presentation and framing of self-improving agents in the paper. The paper often uses the term "self-awareness" without qualification. I'm concerned that anthropomorphization leads to a less scientific discussions and I would recommend the authors try to rewrite the paper to remove phrases like "Our Godel Agent achieves self-awareness".

Moreover, potential risks from self-improving AI are not appropriately discussed. There is a significant literature on risks from self-improving AIs and this is a key concern in the AGI safety community. I think this literature should be acknowledged in the paper and there should be a more complete discussion of broader impacts of this technique than currently the case.

**Questions:**

**Clarification questions**: Please answer the following questions to help me better evaluate the paper.

* Does the learning agent ever see the data the final policy will be evaluated on?
* For how many iterations do you run the Godel agent and how did you determine this hyperparameter?
* In general, how did you tune prompt variations and hyperparameters for this agent? How did you prevent overfitting to the test data?
* Could you please clarify the action space of the agent and how the agent learns about the possible interactions? Is it only given the prompt in Appendix A or is it also given few-shot samples for example?
* How does the agent execute the policy? Is this happening automatically each iteration, or does the agent manually chose to exit the policy? Is this different during iteration and during evaluation?

**Concerns**: Addressing the following concerns would make me reconsider my score.
* Clarify the evaluation protocol.
* Justify the choice of having baselines that use less inference compute. Provide a best-of-n baseline or argue why this is not a good comparison.
* Clarify the comparison to Meta Agent Search and wheather your method produces qualitatively better solutions.
* Justify the choice to use GPT-3.5 for the policy models. If possible, provide some results using GPT-4 for the policies.

**Overall assessment**
I'm intrigued by the premise of the paper, but quite concerned about the evaluation and reproducibility. If the authors can address my concerns and answer my questions, I'm willing to reconsider my score.

---

> ### Author Response · Authors · 2024-11-15
> **Official Response to Reviewer ALKV by Authors -- Part One**
>
> Thanks for your careful and valuable comments. We will explain your concerns point by point.
>
> > Does the learning agent ever see the data the final policy will be evaluated on?
>
> The agent does not have access to the test data. In fact, during the evolving process, we only utilize validation data, as detailed in Appendix B. To ensure a fair comparison with baseline methods, the selection of valid data follows the approach used in the paper[1]. We use the scores on the valid set as feedback from the environment to guide the Gödel Agent in optimizing its policies.
>
> > For how many iterations do you run the Godel agent and how did you determine this hyperparameter?
>
> As stated in line 310 of the paper, we conducted 6 independent self-improvement cycles for each task, with a maximum of 30 iterations per cycle. However, based on our experimental observations, Gödel Agent often stops iterating before reaching this limit. The choice of 30 iterations as a hyperparameter was determined empirically through multiple self-improvement trials. Specifically, in 50 experiments conducted on MGSM, we observed that in 42 cases, Gödel Agent terminated early, deeming its current policy sufficiently optimized before reaching 30 iterations.
>
> > In general, how did you tune prompt variations and hyperparameters for this agent? How did you prevent overfitting to the test data?
>
> Thank you for this question. Due to Gödel Agent's capability for unrestricted self-modification, there was no need for manual tuning of prompts or other hyperparameters. During the self-improvement process, the agent autonomously optimizes these aspects. This self-optimization ability is a key advantage of our approach compared to others. Consequently, the agent does not overfit the test data, as all adjustments are driven internally by the agent based solely on its interactions with the validation set.
>
> > Could you please clarify the action space of the agent and how the agent learns about the possible interactions? Is it only given the prompt in Appendix A or is it also given few-shot samples for example?
>
> The action space of Gödel Agent evolves continuously as it undergoes self-optimization. Initially, as described in Section 3 of the paper, the action space includes: (1) reading its own code, (2) modifying its own code, (3) evaluating its current policy using a utility function, and (4) recursively entering the next level of self-improvement. These actions were deliberately designed to align with the core capabilities of Gödel Agent.
>
> During the self-improvement process, Gödel Agent is free to create new actions and tools, which means its action space is not fixed but rather expands dynamically based on its needs. For example, in our experiments, Gödel Agent autonomously developed tools such as numerical parsers, calculators, and even assistant agents.
>
> Regarding how the agent learns about possible interactions, this is facilitated through the initial prompt provided in Appendix A. The prompt sets up the foundational understanding, encouraging the agent to analyze, explore, and construct new actions. However, we deliberately avoided using few-shot because they could introduce human priors and influence agent’s behavior. Our approach is to minimize human intervention.
>
> > How does the agent execute the policy? Is this happening automatically each iteration, or does the agent manually chose to exit the policy? Is this different during iteration and during evaluation?
>
> Thank you for this insightful question. The execution of the policy is not automatic at every iteration. Instead, testing the policy using the utility function is treated as one of the actions available to the agent. During the iterative self-improvement phase, Gödel Agent autonomously decides whether or not to test its current policy based on its assessment. It does not execute the policy at every step but rather chooses when it is optimal to do so.
>
> In contrast, during the final evaluation phase, the process is different: we manually invoke the utility function to test the agent’s final policy. This ensures that the evaluation is consistent and unbiased. Thus, the approach during iteration—where the agent makes autonomous decisions—is distinct from the controlled, manual process used during the final evaluation.
>
> > Clarify the evaluation protocol
>
> As mentioned earlier, each task involves a utility function that helps evaluate the agent’s policy. During the iterative process, Gödel Agent decides autonomously when to invoke this utility function to test its current policy. In the final evaluation phase, we manually test the final policy using the same utility function.
>
> A key distinction between iteration and evaluation is the data used: during the iterative process, the utility function is applied to the **valida data** to guide the agent's self-improvement. In contrast, during the evaluation phase, we use **test data** to assess the performance of the agent’s final policy.

---

> ### Author Response · Authors · 2024-11-15
> **Official Response to Reviewer ALKV by Authors -- Part Two**
>
> > Justify the choice of having baselines that use less inference compute. Provide a best-of-n baseline or argue why this is not a good comparison.
>
> Thank you for your question. It seems there may be a misunderstanding about the computation cost of our method. During the optimization iterations, Gödel Agent does not generate a new strategy or test it at every step. Instead, it selectively optimizes its policy at certain steps and evaluates it based on feedback from the validation set. The specific steps where this happens are determined autonomously by the agent.
>
> In terms of inference computation, the cost is similar to that of the baselines because we only test the final optimized policy once. Therefore, the additional computational cost is primarily associated with the optimization process, where Gödel Agent uses this compute to replace the human labor typically involved in manually designing and refining agent strategies. Moreover, for each task, only a one-time optimization process is required, and the generated policy can always be applied on that task, so we consider this consumption acceptable.
>
> The baseline methods, on the other hand, are manually designed by human experts, which consumes significant time and resources. Thus, while Gödel Agent uses more compute during optimization, it eliminates the need for manual design, which is far more resource-intensive.
>
> > Clarify the comparison to Meta Agent Search and wheather your method produces qualitatively better solutions.
>
> Thank you for the opportunity to clarify the comparison between our method and Meta Agent Search (MAS). In MAS, a fixed meta-agent is manually designed to create task-specific agents by interacting with the environment. However, this meta-agent remains static throughout the process, meaning the final optimized results are inherently constrained by the initial design and capabilities of the meta-agent.
>
> In contrast, our approach, inspired by the Gödel machine, moves beyond the meta-learning paradigm by endowing the agent with self-referential capabilities. This allows the agent to evolve not only its task-specific part but also its own optimization part. As a result, both the agent's ability to optimize and its performance on downstream tasks improve simultaneously, leading to more efficient and effective solutions.
>
> A clear example of this qualitative improvement can be seen in the "Game of 24" task as shown in Appendix C.2. Gödel Agent was able to design a search-based algorithm that solves the task with 100% accuracy, whereas MAS is limited to searching within predefined spaces and designing prompt-based methods. This shows that it can generate solutions that are not only more flexible but also fundamentally more capable than those produced by a fixed meta-agent.
>
> > Justify the choice to use GPT-3.5 for the policy models. If possible, provide some results using GPT-4 for the policies.
>
> We chose to use GPT-4 only during the learning phase because it offers superior reasoning capabilities, which are essential for the self-improvement process where the agent generates and refines its strategies. This phase involves replacing human labor with autonomous optimization, and GPT-4's enhanced reasoning helps Gödel Agent improve its policy efficiently.
>
> For task execution, we opted for GPT-3.5 for all methods because, in real-world applications, executing policies often requires frequent API calls, and GPT-3.5 is significantly more cost-effective for this purpose.
>
> The policy generated by Gödel Agent is fixed after the learning phase, much like CoT, where the learned policy remains unchanged during execution. Therefore, the decision to use GPT-3.5 for execution maintains fairness in comparison.
>
> > The paper often uses the term "self-awareness" without qualification. I'm concerned that anthropomorphization leads to a less scientific discussions and I would recommend the authors try to rewrite the paper to remove phrases like "Our Godel Agent achieves self-awareness".
>
> Thank you for your thoughtful comment and suggestion. We apologize for any confusion caused by our use of the term "self-awareness." In this context, we intended to convey that Gödel Agent has the capability to introspect and read its own code and files, not to imply any philosophical sense of consciousness or awareness. We have emphasized this point more clearly in the revised version of the paper. If you find that the current explanation is still not sufficiently clear, would using the term of *self-introspection* be more appropriate?
>
> We hope that our responses have addressed your questions. We also hope that you find our approach exciting, as it represents a fundamental departure from previous methods. If you have any further questions, please feel free to reach out, and we will be happy to address them. Thank you once again for your valuable feedback.
>
> [1] Automated Design of Agentic Systems

---

> > ### Comment · Reviewer_ALKV · 2024-11-20
> >
> > Thanks for the detailed response. This clarified a couple of concerns, and I will consider updating my review. In the meantime, I have a few follow-up questions:
> >
> > > there was no need for manual tuning of prompts or other hyperparameters
> >
> > I was thinking primarily about the _initial_ prompting and scaffolding. How much did you find performance to rely on the initialization and how did you iterate and choose the initialization?
> >
> > > The policy generated by Gödel Agent is fixed after the learning phase, much like CoT, where the learned policy remains unchanged during execution. Therefore, the decision to use GPT-3.5 for execution maintains fairness in comparison.
> >
> > My concern is that by using GPT-4 during the learning phase, you indirectly expose the final policy to GPT-4 level capabilities. I think it's not fair to compare this solely to human work during prompt design, because human's didn't design the prompts to optimize performance on individual tasks like the Godel agent does in your setup. To run a proper human baseline, you'd have to have humans do that explicitly for these tasks. Lacking that, I think it is not appropriate to use GPT-3.5 for running the final policy.
> >
> > I think my overall concern here is what is the core claim about the method? Is the claim that it performs better than humans at designing prompts? In that case a human baseline would be necessary to understand the results. Or is the claim that the Godel agent performs better than a set of general scaffolding? In that case I think the compute should be matched or at least evaluate separately because we do not actually spend human effort to design new scaffolding for each task. Could you please clarify the key empirical claim and how your experimental evidence supports it?
> >
> > > If you find that the current explanation is still not sufficiently clear, would using the term of self-introspection be more appropriate?
> >
> > Yes I think introspection would be a more appropriate term, though it would still need some explanation.

---

> > > ### Author Response · Authors · 2024-11-20
> > >
> > > > I was thinking primarily about the *initial* prompting and scaffolding. How much did you find performance to rely on the initialization and how did you iterate and choose the initialization?
> > >
> > > The initial prompt is detailed in Appendix A. The initial prompt was structured around three key components: Core Capabilities, Core Methods and Guiding Principles of Gödel Agent.
> > >
> > > The scaffolding was similarly minimal, consisting of a process where the initial prompt is passed to the LLM, which then generates an action sequence (e.g., reading its own code, modifying its code, reading/writing files, running code snippets). These actions are executed iteratively and then continue to self-improve recursively.
> > >
> > > In our initial experiments, we found Gödel Agent’s performance was not sensitive to the initialization. Therefore, we adhered to a simplified approach, ensuring the prompt was clear and concise while equipping the scaffolding with only the basic recursive and functional capabilities required.
> > >
> > > > I think my overall concern here is what is the core claim about the method? Is the claim that it performs better than humans at designing prompts? In that case a human baseline would be necessary to understand the results. Or is the claim that the Godel agent performs better than a set of general scaffolding? In that case I think the compute should be matched or at least evaluate separately because we do not actually spend human effort to design new scaffolding for each task. Could you please clarify the key empirical claim and how your experimental evidence supports it?
> > >
> > > The core claim of our work is that Gödel Agent, as a self-referential framework capable of recursive self-improvement, outperforms both human-designed fixed agents and meta-learning optimized agents. This is a paradigm shift: from directly using existing agent frameworks, to using meta-learning algorithms to optimize and obtain an agent framework, and then to allowing agent frameworks to recursively self-improve.
> > >
> > > **Comparison with Human-Designed Agents**: Fixed human-designed agents typically underperform on new tasks due to their limited transferability. To achieve better performance, these agents often require significant human effort for task-specific redesign. Gödel Agent, by contrast, autonomously learns effective strategies without requiring human-driven optimization. While we acknowledge the possibility that after task-specific optimization by experts, human-designed agents could surpass Gödel Agent, the purpose of Gödel Agent is to reduce this human labor by leveraging GPT-4 to independently explore and learn.
> > >
> > > **Comparison with Meta-Learning Optimized Agents**: Meta-learning optimized agents can adapt to specific tasks (e.g., using GPT-4 for optimization), but their meta-learning algorithms are fixed and cannot be improved further. The comparison with meta-learning agents is fair, as both Gödel Agent and ADAS use the same underlying models and compute resources during optimization.
> > >
> > > > Yes I think introspection would be a more appropriate term, though it would still need some explanation.
> > >
> > > Thank you for the suggestion. We will use *self-introspection* in our next version. By self-introspection, we mean Gödel Agent’s capability to introspectively analyze its own structure, including reading and understanding its own code and files.

---

> > > > ### Comment · Reviewer_ALKV · 2024-11-22
> > > >
> > > > Thanks for the response. I've considered your response, and read through the other reviews and responses. Some of my concerns have been addressed, and I now think the evaluation as presented is broadly solid. Consequently, I will increase my score from 3 to 5. However, I remain unsure about the evaluation setup and baselines as well as the overall readability of the paper.

---

> > > > > ### Author Response · Authors · 2024-11-23
> > > > >
> > > > > Thank you for updating your evaluation. We appreciate your constructive feedback and are glad to hear that some of your concerns have been addressed.
> > > > >
> > > > > We’ve also revised the paper to improve clarity and addressed the questions other reviewers proposed. Could you let us know if any specific questions or issues remain that we can clarify further?
> > > > >
> > > > > Thank you again for your time and support!

---

> > > > > ### Author Response · Authors · 2024-12-02
> > > > > **Official Comment to Reviewer ALKV by Authors**
> > > > >
> > > > > In our previous responses, we have clarified the evaluation setup and the rationale behind the choice of baselines. Additionally, we have revised the paper based on the feedback provided by all reviewers to improve its overall clarity and presentation.
> > > > >
> > > > > If there are any specific concerns or areas of the evaluation setup, baselines, or readability that remain unclear, we would greatly appreciate it if you could point them out. We will do our best to address them thoroughly.

---

### Author Response · Authors · 2024-11-22
**Follow-up: request for engagement during discussion period**

Dear Reviewers,

With only a few days remaining in the discussion period, we would greatly appreciate your engagement to ensure a constructive dialogue. In our revision, we’ve worked hard to address your feedback, making significant improvements to the paper. Detailed responses to reviewer-specific points in the each separate comment below.

We are eager to hear your thoughts on these updates and hope you’ll have a chance to review our responses. We value your time and effort in shaping this submission.

Thank you again for your thoughtful reviews and for considering our responses.

Best regards,

The Authors of Submission 13979

---

### Meta-Review · Area_Chair_STvU · 2024-12-19

**Metareview:**

This paper proposes an ambitious goal of creating a fully self-improving agent that can modify its own programmatic code. However, there are a few concerns that make this work not yet ready for publication. Firstly, the evaluation is not fair on a compute basis, since there is no control on the number of generation calls (mentioned by Reviewer ALKV). Secondly, the reported results for the Gödel agent are not better than ADAS, as the error bars are all overlapping, with the exception of MGSM. Deeper analysis revealing how the incremental program modifications and final programs produced by Gödel agent differ from and contribute to its success with respect to baselines like ADAS would make the contribution here much clearer.

**Additional Comments On Reviewer Discussion:**

The reviewers generally felt this work did not provide enough details around how the method works in practice. Moreover, reviewers found the evaluations did not provide enough understanding for whether the proposed method indeed improves over existing methods, especially when controlling for compute cost.

---

### Decision · Program_Chairs · 2025-01-22

Reject